# Adaptive Linear Estimating Equations

**Mufang Ying**
Department of Statistics
Rutgers University - New Brunswick
my426@scarletmail.rutgers.edu

**Koulik Khamaru**
Department of Statistics
Rutgers University - New Brunswick
k1241@stat.rutgers.edu

**Cun-Hui Zhang**
Department of Statistics
Rutgers University - New Brunswick
czhang@stat.rutgers.edu

## Abstract

Sequential data collection has emerged as a widely adopted technique for enhancing the efficiency of data gathering processes. Despite its advantages, such data collection mechanism often introduces complexities to the statistical inference procedure. For instance, the ordinary least squares (OLS) estimator in an adaptive linear regression model can exhibit non-normal asymptotic behavior, posing challenges for accurate inference and interpretation. In this paper, we propose a general method for constructing debiased estimator which remedies this issue. It makes use of the idea of adaptive linear estimating equations, and we establish theoretical guarantees of asymptotic normality, supplemented by discussions on achieving near-optimal asymptotic variance. A salient feature of our estimator is that in the context of multi-armed bandits, our estimator retains the non-asymptotic performance of the least squares estimator while obtaining asymptotic normality property. Consequently, this work helps connect two fruitful paradigms of adaptive inference: a) non-asymptotic inference using concentration inequalities and b) asymptotic inference via asymptotic normality.

## 1 Introduction

Adaptive data collection arises as a common practice in various scenarios, with a notable example being the use of (contextual) bandit algorithms. Algorithms like these aid in striking a balance between exploration and exploitation trade-offs within decision-making processes, encompassing domains such as personalized healthcare and web-based services [35, 24, 3, 22]. For instance, in personalized healthcare, the primary objective is to choose the most effective treatment for each patient based on their individual characteristics, such as medical history, genetic profile, and living environment. Bandit algorithms can be used to allocate treatments based on observed response, and the algorithm updates its probability distribution to incorporate new information as patients receive treatment and their response is observed. Over time, the algorithm can learn which treatments are the most effective for different types of patients.

Although the adaptivity in data collection improves the quality of data, the sequential nature (non-iid) of the data makes the inference procedure quite challenging [34, 26, 5, 28, 27, 10, 30, 29]. There is a lengthy literature on the problem of parameter estimation in the adaptive design setting. In a series of work [15, 19, 17], the authors studied the consistency of the least squares estimator for an adaptive

37th Conference on Neural Information Processing Systems (NeurIPS 2023).

linear model. In a later work, Lai [14] studied the consistency of the least squares estimator in a nonlinear regression model. The collective wisdom of these papers is that, for adaptive data collection methods, standard estimators are consistent under a mild condition on the maximum and minimum eigenvalues of the covariance matrix [19, 14]. In a more recent line of work [1, 2], the authors provide a high probability upper bound on the $\ell_2$-error of the least squares estimator for a linear model. We point out that, while the high probability bounds provide a quantitative understanding of OLS, these results assume a stronger sub-Gaussian assumption on the noise variables.

The problem of inference, i.e. constructing valid confidence intervals, with adaptively collected data is much more delicate. Lai and Wei [19] demonstrated that for a unit root autoregressive model, which is an example of adaptive linear regression models, the least squares estimator doesn't achieve asymptotic normality. Furthermore, the authors showed that for a linear regression model, the least squares estimator is asymptotically normal when the data collection procedure satisfies a stability condition. Concretely, letting $\boldsymbol{x}_i$ denote the covariate associated with $i$-th sample, the authors require

$$\mathbf{B}_n^{-1}\mathbf{S}_n \xrightarrow{p} \mathbf{I} \tag{1}$$

where $\mathbf{S}_n = \sum_{i=1}^n \boldsymbol{x}_i \boldsymbol{x}_i^\top$ and $\{\mathbf{B}_n\}_{n \geq 1}$ is a sequence of *non-random* positive definite matrices. Unfortunately, in many scenarios, the stability condition (1) is violated [38, 19]. Moreover, in practice, it might be difficult to verify whether the stability condition (1) holds or not. In another line of research [10, 36, 37, 4, 9, 28, 31, 25, 38], the authors assume knowledge of the underlying data collection algorithm and provide asymptotically valid confidence intervals. While this approach offers intervals under a much weaker assumption on the underlying model, full knowledge of the data collection algorithm is often unavailable in practice.

**Online debiasing based methods:** In order to produce valid statistical inference when the stability condition (1) does not hold, some authors [8, 7, 13] utilize the idea of online debiasing. At a high level, the online debiased estimator reduces bias from an initial estimate (usually the least squares estimate) by adding some correction terms, and the online debiasing procedure does not require the knowledge of the data generating process. Although this procedure guarantees asymptotic reduction of bias to zero, the bias term's convergence rate can be quite slow.

In this work, we consider estimating the unknown parameter in an adaptive linear model by using a set of adaptive linear estimating equations (ALEE). We show that our proposed ALEE estimator achieves asymptotic normality without knowing the exact data collection algorithm while addressing the slowly decaying bias problem in online debiasing procedure.

## 2 Background and problem set-up

In this section, we provide the background for our problem and set up a few notations. We begin by defining the adaptive data collection mechanism for linear models.

### 2.1 Adaptive linear model

Suppose a scalar response variable $y_t$ is linked to a covariate vector $\boldsymbol{x}_t \in \mathbb{R}^d$ at time $t$ via the linear model:

$$y_t = \boldsymbol{x}_t^\top \boldsymbol{\theta}^* + \epsilon_t \qquad \text{for } t \in [n], \tag{2}$$

where $\boldsymbol{\theta}^* \in \mathbb{R}^d$ is the unknown parameter of interest.

In an adaptive linear model, the regressor $\boldsymbol{x}_t$ at time $t$ is assumed to be a (unknown) function of the prior data point $\{\boldsymbol{x}_1, y_1, \ldots, \boldsymbol{x}_{t-1}, y_{t-1}\}$ as well as additional source of randomness that may be present in the data collection process. Formally, we assume there is an increasing sequence of $\sigma$-fields $\{\mathcal{F}_t\}_{t \geq 0}$ such that

$$\sigma(\boldsymbol{x}_1, y_1, \ldots, \boldsymbol{x}_{t-1}, y_{t-1}, \boldsymbol{x}_t) \in \mathcal{F}_{t-1} \qquad \text{for } t \in [n].$$

For the noise variables $\{\epsilon_t\}_{t \geq 1}$ appearing in equation (2), we impose the following conditions

$$\mathbb{E}[\epsilon_t | \mathcal{F}_{t-1}] = 0, \quad \mathbb{E}[\epsilon_t^2 | \mathcal{F}_{t-1}] = \sigma^2, \quad \text{and} \quad \sup_{t \geq 1} \mathbb{E}[|\epsilon_t/\sigma|^{2+\delta} | \mathcal{F}_{t-1}] < \infty, \tag{3}$$

for some $\delta > 0$. The above condition is relatively mild compared to a sub-Gaussian condition.

Examples of adaptive linear model arise in various problems, including multi-armed and contextual bandit problems, dynamical input-output systems, adaptive approximation schemes and time series models. For instance, in the context of the multi-armed bandit problem, the design vector $\boldsymbol{x}_t$ is one of the basis vectors $\{\boldsymbol{e}_k\}_{k \in [d]}$, representing an arm being pulled, while $\boldsymbol{\theta}^*$, $y_t$ represent the true mean reward vector and reward at time $t$, respectively.

## 2.2 Adaptive linear estimating equations

As we mentioned earlier, the OLS estimator can fail to achieve asymptotic normality due to the instability of the covariance matrix with adaptively collected data. To get around this issue, we consider a different approach ALEE (adaptive linear estimating equations). Namely, we obtain an estimate by solving a system of linear estimating equations with adaptive weights,

$$\text{ALEE:} \qquad \sum_{t=1}^{n} \boldsymbol{w}_t(y_t - \boldsymbol{x}_t^\top \widehat{\boldsymbol{\theta}}_{\text{ALEE}}) = \mathbf{0}. \tag{4}$$

Here the weight $\boldsymbol{w}_t \in \mathbb{R}^d$ is chosen in a way that $\boldsymbol{w}_t \in \mathcal{F}_{t-1}$ for $t \in [n]$. Let us now try to gain some intuition behind the construction of ALEE. Rewriting equation (4), we have

$$\{\textstyle\sum_{t=1}^{n} \boldsymbol{w}_t \boldsymbol{x}_t\} \cdot (\widehat{\boldsymbol{\theta}}_{\text{ALEE}} - \boldsymbol{\theta}^*) = \sum_{t=1}^{n} \boldsymbol{w}_t \epsilon_t. \tag{5}$$

Notably, the choice of $\boldsymbol{w}_t \in \mathcal{F}_{t-1}$ makes $\sum_{t=1}^{n} \boldsymbol{w}_t \epsilon_t$ the sum of a martingale difference sequence. Our first theorem postulates conditions on the weight vectors $\{\boldsymbol{w}_t\}_{t \geq 1}$ such that the right-hand side of (5) converges to normal distribution asymptotically. Throughout the paper, we use the shorthand $\mathbf{W}_t = (\boldsymbol{w}_1, \ldots, \boldsymbol{w}_t)^\top \in \mathbb{R}^{t \times d}$, $\mathbf{X}_t = (\boldsymbol{x}_1, \ldots, \boldsymbol{x}_t)^\top \in \mathbb{R}^{t \times d}$.

**Proposition 2.1.** *Suppose condition* (3) *holds and the predictable sequence* $\{\boldsymbol{w}_t\}_{1 \leq t \leq n}$ *satisfies*

$$\max_{1 \leq t \leq n} \|\boldsymbol{w}_t\|_2 = o_p(1) \qquad and \qquad \left\|\mathbf{I}_d - \mathbf{W}_n^\top \mathbf{W}_n\right\|_{\text{op}} = o_p(1). \tag{6}$$

*Let* $\mathbf{A}_w = \mathbf{V}_w \mathbf{U}_w^\top \mathbf{X}_n$ *with* $\mathbf{W}_n = \mathbf{U}_w \boldsymbol{\Lambda}_w \mathbf{V}_w^\top$ *being the SVD of* $\mathbf{W}_n$. *Then,*

$$\mathbf{A}_w(\widehat{\boldsymbol{\theta}}_{\text{ALEE}} - \boldsymbol{\theta}^*)/\widehat{\sigma} \xrightarrow{d} \mathcal{N}(\mathbf{0}, \mathbf{I}_d), \tag{7}$$

*where* $\widehat{\sigma}$ *is any consistent estimator for* $\sigma$.

**Proof.** Invoking the second part of the condition (6), we have that $\boldsymbol{\Lambda}_w$ is invertible for large $n$, and $\|\mathbf{V}_w \boldsymbol{\Lambda}_w^{-1} \mathbf{V}_w^\top - \mathbf{I}_d\|_{\text{op}} = o_p(1)$. Utilizing the expression (5), we have

$$\mathbf{A}_w(\widehat{\boldsymbol{\theta}}_{\text{ALEE}} - \boldsymbol{\theta}^*)/\sigma = \mathbf{V}_w \boldsymbol{\Lambda}_w^{-1} \mathbf{V}_w^\top \mathbf{W}_n^\top \mathbf{X}_n (\widehat{\boldsymbol{\theta}}_{\text{ALEE}} - \boldsymbol{\theta}^*)/\sigma = \mathbf{V}_w \boldsymbol{\Lambda}_w^{-1} \mathbf{V}_w^\top \sum_{t=1}^{n} \boldsymbol{w}_t \epsilon_t/\sigma.$$

Invoking the stability condition on the weights $\{\boldsymbol{w}_t\}$ and using the fact that $\sum_{t=1}^{n} \boldsymbol{w}_t \epsilon_t$ is a martingale difference sequence, we conclude from martingale central limit theorem [11, Theorem 2.1] that

$$\textstyle\sum_{t=1}^{n} \boldsymbol{w}_t \epsilon_t/\sigma \xrightarrow{d} \mathcal{N}(\mathbf{0}, \mathbf{I}_d).$$

Combining the last equation with $\|\mathbf{V}_w \boldsymbol{\Lambda}_w^{-1} \mathbf{V}_w^\top - \mathbf{I}_d\|_{\text{op}} = o_p(1)$ and using Slutsky's theorem yield

$$\mathbf{A}_w(\widehat{\boldsymbol{\theta}}_{\text{ALEE}} - \boldsymbol{\theta}^*)/\sigma \xrightarrow{d} \mathcal{N}(\mathbf{0}, \mathbf{I}_d).$$

The claim of Proposition 2.1 now follows from Slutsky's theorem.

A few comments regarding the Proposition 2.1 are in order. Straightforward calculation shows

$$\mathbf{A}_w^\top \mathbf{A}_w = \mathbf{X}_n^\top \mathbf{P}_w \mathbf{X}_n \preceq \mathbf{S}_n, \qquad \text{where} \qquad \mathbf{P}_w = \mathbf{W}_n(\mathbf{W}_n^\top \mathbf{W}_n)^{-1} \mathbf{W}_n^\top. \tag{8}$$

In words, the volume of the confidence region based on (7) is always larger than the confidence region generated by the least squares estimate. Therefore, the ALEE-based inference, which is consistently valid, exhibits a reduced efficiency in cases where both types of confidence regions are valid. Compared with the confidence regions based on OLS, the advantage of the ALEE approach is to provide flexibility in the choice of weights to guarantee the validity of the CLT conditions (6).

Next, note that the matrix $\mathbf{A}_w$ is asymptotically equivalent to the matrix $\mathbf{W}_n^\top \mathbf{X}_n$ (see equation (5)) under the stability condition (6). The benefit of this reformulation is that it helps us better understand

efficiency of ALEE compared with the OLS. This has led us to define a notion of *affinity* between the weights $\{w_t\}_{t \geq 1}$ and covariates $\{x_t\}_{t \geq 1}$ for better understanding of the efficiency of ALEE and ways to design nearly optimal weights, as it will be clear in the next section.

Finally, it is straightforward to obtain a consistent estimate for $\sigma$. For instance, assuming $\log(\lambda_{\max}(\mathbf{X}_n^\top \mathbf{X}_n))/n \xrightarrow{a.s.} 0$ and the noise condition (3), we have

$$\widehat{\sigma}^2 := \frac{1}{n} \sum_{t=1}^n (y_t - x_t^\top \widehat{\theta}_{\mathrm{LS}})^2 \xrightarrow{a.s.} \sigma^2. \tag{9}$$

Here, $\widehat{\theta}_{\mathrm{LS}}$ refers to the least squares estimate. See [19, Lemma 3] for a detailed proof of equation (9).

## 3 Main results

In this section, we propose methods to construct weights $\{w_t\}_{t \geq 1}$ which satisfy the stability property (6), and study the resulting ALEE. Section 3.1 is devoted to the multi-armed bandit case, Section 3.2 to an autoregressive model, and Section 3.3 to the contextual bandit case. Before delving into details, let us try to understand intuitively how to construct weights that have desirable properties.

The expression (8) reveals that the efficiency of ALEE depends on the projection of the data matrix $\mathbf{X}_n$ on $\mathbf{W}_n$. Thus, the efficiency of the ALEE approach can be measured by the principal angles between the random projections $\mathbf{P}_w$ in (8) and $\mathbf{P}_x = \mathbf{X}_n \mathbf{S}_n^{-1} \mathbf{X}_n^\top$. Accordingly, we define the *affinity* $\mathcal{A}(\mathbf{W}_n, \mathbf{X}_n)$ of the weights $\{w_t\}_{t \geq 1}$ as the cosine of the largest principle angle, or equivalently

$$\mathcal{A}(\mathbf{W}_n, \mathbf{X}_n) = \sigma_d(\mathbf{P}_w \mathbf{P}_x) = \sigma_d\big(\mathbf{U}_w^\top \mathbf{X}_n \mathbf{S}_n^{-1/2}\big) \tag{10}$$

as the $d$-th largest singular value of $\mathbf{P}_w \mathbf{P}_x$. Formally, the above definition captures the cosine of the angle between the two subspaces spanned by the columns of $\mathbf{X}_n$ and $\mathbf{W}_n$, respectively [12]. Good weights $\{w_t\}_{t \geq 1}$ are those with relatively large affinity or

$$\mathbf{U}_w \propto \mathbf{X}_n \mathbf{S}_n^{-1/2} \qquad \text{(approximately).} \tag{11}$$

### 3.1 Multi-armed bandits

In the context of the $K$-arm bandit problem, the Gram matrix has a diagonal structure, which means that we can focus on constructing weights $\{w_t\}_{t \geq 1}$ for each coordinate independently. For an arm $k \in [K]$ and round $t \geq 1$, define

$$s_{t,k} = s_0 + \sum_{i=1}^t x_{i,k}^2 \qquad \text{for some positive } s_0 \in \mathcal{F}_0. \tag{12}$$

Define the $k$-th coordinate of the weight $w_t$ as

$$w_{t,k} = f\left(\frac{s_{t,k}}{s_0}\right) \cdot \frac{x_{t,k}}{\sqrt{s_0}} \quad \text{with} \quad f(x) = \sqrt{\frac{\log 2}{x \cdot \log(e^2 x) \cdot (\log\log(e^2 x))^2}}. \tag{13}$$

The intuition behind the above construction is as follows. The discussion at near equation (11) indicates that the $k$-th coordinate of $w_t$ should be proportional to $x_{t,k}/(\sum_{i \leq n} x_{i,k}^2)^{1/2}$. However, the weight $w_t$ is required to be predictable, which can only depend on the data points [1] up to time $t$. Consequently, we approximate the sum $\sum_{i \leq n} x_{i,k}^2$ by the partial sum $s_{t,k}$ in (12). Finally, note that

$$w_{t,k} = f\left(\frac{s_{t,k}}{s_0}\right) \cdot \frac{x_{t,k}}{\sqrt{s_0}} \approx \frac{x_{t,k}}{\sqrt{s_{t,k}}}. \tag{14}$$

The logarithmic factors in (13) ensure that the stability conditions (6) hold. In the following theorem, we generalize the above method as a general strategy for constructing weights $\{w_t\}_{t \geq 1}$ satisfying the stability condition (6).

---

[1]Note that $x_{t,k} \in \mathcal{F}_{t-1}$ can be used to construct $w_t$

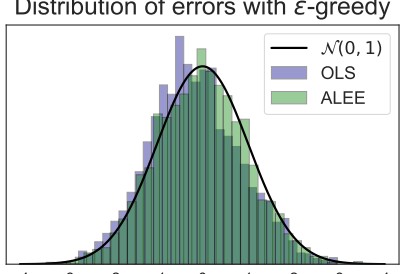

Figure 1: Empirical distribution of the standardized estimation errors from OLS and ALEE approach. Results are obtained with a dataset of size $n = 1000$ and 3000 independent replications. Left: AR(1) model $y_t = y_{t-1} + \epsilon_t$ with independent errors $\epsilon_t \sim \mathcal{N}(0, 1)$. Right: Two-armed bandit problem with equal arm mean $\theta_1^* = \theta_2^* = 0.3$ and independent noise $\epsilon_t \sim \mathcal{N}(0, 1)$. Figure 4 in Section C.3 considers the same setting with centered Poisson noise, which is not sub-Gaussian.

### 3.1.1 Stable weight construction strategy

Consider a positive decreasing function $f(x)$ on the interval $[1, \infty)$ with an increasing derivative $f'(x)$. Let $f$ satisfy the conditions: $f'/f$ is increasing,

$$\int_1^\infty f^2(x)dx = 1, \quad \text{and} \quad \int_1^\infty f(x)dx = \infty. \tag{15}$$

With $s_0 \in \mathcal{F}_0$, we define weight $w_{t,k}$ as

$$w_{t,k} = f\left(\frac{s_{t,k}}{s_0}\right)\frac{x_{t,k}}{\sqrt{s_0}} \quad \text{with} \quad s_{t,k} = s_0 + \sum_{i=1}^t x_{i,k}^2. \tag{16}$$

A key condition that ensures the weights $\{w_{t,k}\}_{t \geq 1}$ satisfy the desirable stability property (6) is

$$\max_{1 \leq t \leq n} f^2\left(\frac{s_{t,k}}{s_0}\right)\frac{x_{t,k}^2}{s_0} + \max_{1 \leq t \leq n}\left(1 - \frac{f(s_{t,k}/s_0)}{f(s_{t-1,k}/s_0)}\right) + \int_{s_{n,k}/s_0}^\infty f^2(x)dx = o_p(1). \tag{17}$$

For multi-armed bandits, this condition is automatically satisfied when both quantities $1/s_0$ and $s_0/s_{n,k}$ converge to zero in probability. Putting together the pieces, we have the following result for multi-armed bandits.

**Theorem 3.1.** *Suppose condition (3) holds and $1/s_0 + s_0/s_{n,k} = o_p(1)$ for some $k \in [K]$. Then, the $k$-th coordinate $\widehat{\theta}_{\text{ALEE},k}$, obtained using weights from equation (16), satisfies*

$$\left(\widehat{\theta}_{\text{ALEE},k} - \theta_k^*\right) \cdot \int_1^{s_{n,k}/s_0} \frac{\sqrt{s_0}}{\widehat{\sigma}}f(x)dx \xrightarrow{d} \mathcal{N}(0, 1), \tag{18}$$

*where $\widehat{\sigma}$ is a consistent estimate of $\sigma$. Equivalently,*

$$\frac{\left(\widehat{\theta}_{\text{ALEE},k} - \theta_k^*\right)}{\widehat{\sigma}\sqrt{\sum_{1 \leq t \leq n} w_{t,k}^2}} \cdot \left(\sum_{t=1}^n w_{t,k}x_{t,k}\right) \xrightarrow{d} \mathcal{N}(0, 1). \tag{19}$$

The proof of Theorem 3.1 can be found in Section A.1 of the Appendix. A few comments regarding Theorem 3.1 are in order.

First, the above theorem enables us to construct valid CI in the estimation of the mean $\theta_k^*$ for a sub-optimal arm $k$ when employing an asymptotically optimal allocation rule to achieve the optimal regret in [18] with sample size $\sum_{t \leq n} x_{t,k} \asymp \log n$, or when using a sub-optimal rule to achieve polylog($n$). On the other hand, the classical martingale CLT is applicable to the optimal arm (if unique) under such asymptotically optimal or sub-optimal allocation rules. Consequently, one may obtain a valid CI for the optimal arm from the standard OLS estimate [19]. However, it is important to note that such CIs are not guaranteed for sub-optimal arms.

Next, while Theorem 3.1 holds for any $s_0$ diverging to infinity but of smaller order than $s_{n,k}$ ( which may depend on $k$), the convergence rate of $\sum_{1 \leq t \leq n} w_{t,k} \epsilon_t$ to normality is enhanced by choosing a large value for $s_0$. In practical terms, it is advisable to choose an $s_0$ that is slightly smaller than the best-known lower bound for $s_{n,k}$.

Finally, the choice of function $f$ determines the efficiency of ALEE estimator. For instance, taking function $f(x) = 1/x$, we obtain an estimator with asymptotic variance of order $1/\{s_0 \log^2(s_{n,k}/s_0)\}$, which is only better than what one would get using stopping time results by a logarithmic factor. In the next Corollary, an improved choice of $f$ yields near optimal variance up to logarithmic terms.

**Corollary 3.2.** *Consider the same set of assumptions as stated in Theorem 3.1. The ALEE estimator $\widehat{\theta}_{\mathrm{ALEE},k}$, obtained by using $f(x) = (\beta \log^\beta 2)^{1/2}\{x(\log e^2 x)(\log \log e^2 x)^{1+\beta}\}^{-1/2}$ for any $\beta > 0$ satisfies*

$$\sqrt{\frac{4\beta(\log 2)^\beta}{\log(s_{n,k}/s_0)\{\log \log(s_{n,k}/s_0)\}^{1+\beta}}} \cdot \frac{\sqrt{s_{n,k}}(\widehat{\theta}_{\mathrm{ALEE},k} - \theta_k^*)}{\widehat{\sigma}} \xrightarrow{d} \mathcal{N}(0,1).$$

The proof of this corollary follows directly from Theorem 3.1. For $s_0 = \log n / \log \log n$ in multi-armed bandits with asymptotically optimal allocations, $\log(s_{n,k}/s_0) = (1 + o(1)) \log \log s_{n,k}$.

### 3.1.2 Finite sample bounds for ALEE estimators

One may also construct finite sample confidence intervals for each arm via applying concentration bounds. Indeed, for any arm $k \in K$, we have

$$\{\sum_{t=1}^n w_{t,k} x_{t,k}\} \cdot (\widehat{\theta}_{\mathrm{ALEE},k} - \theta_k^*) = \sum_{t=1}^n w_{t,k} \epsilon_t. \tag{20}$$

Following the construction of $w_{t,k} \in \mathcal{F}_{t-1}$, the term $\sum_{t=1}^n w_{t,k} \epsilon_t$ is amenable to concentration inequalities if we assume that the noise $\epsilon_t$ is sub-Gaussian conditioned on $\mathcal{F}_{t-1}$, i.e.

$$\forall \lambda \in \mathbb{R} \qquad \mathbb{E}[e^{\lambda \epsilon_t} \mid \mathcal{F}_{t-1}] \leq e^{\sigma_g^2 \lambda^2 / 2}. \tag{21}$$

**Corollary 3.3** (Theorem 1 in [1])**.** *Suppose the sub-Gaussian noise condition (21) is in force. Then for any $\delta > 0$ and $\lambda_0 > 0$, the following bound holds with probability at least $1 - \delta$*

$$\left| \sum_{t=1}^n w_{t,k} x_{t,k} \right| \cdot |\widehat{\theta}_{\mathrm{ALEE},k} - \theta_k^*| \leq \sigma_g \sqrt{(\lambda_0 + \sum_{t=1}^n w_{t,k}^2) \cdot \log\left(\frac{\lambda_0 + \sum_{t=1}^n w_{t,k}^2}{\delta^2 \lambda_0}\right)}. \tag{22}$$

**Remark 3.4.** *In the context of multi-armed bandit, by considering the function $f$ in Corollary 3.2 with $\beta = 1$ and Corollary 3.3 with $\lambda_0 = 1$, we derive that with probability at least $1 - \delta$*

$$|\widehat{\theta}_{\mathrm{ALEE},k} - \theta_k^*| \leq \sigma_g \sqrt{\log(2/\delta^2)} \frac{\sqrt{2 + \log(s_{n,k}/s_0)} \log\{2 + \log(s_{n,k}/s_0)\}}{\sqrt{s_{n,k}} - \sqrt{s_0}} \tag{23}$$

*provided $s_0 > 1$. See Section A.2 of the Appendix for a proof of this argument. Recall that $\sqrt{s_{n,k}} = (s_0 + \sum_{i \leq n} x_{i,k}^2)^{1/2}$, the bound is in the same spirit as existing finite sample bounds for the OLS estimator for arm means [1, 21]. In simple terms, the ALEE estimator behaves similarly to the OLS estimator in a non-asymptotic setting while still maintaining asymptotic normality.*

## 3.2 Autoregressive time series

Next, we focus on an autoregressive time series model

$$y_t = \theta^* y_{t-1} + \epsilon_t \quad \text{for } t \in [n], \tag{24}$$

where $y_0 = 0$. Note that the above model is a special case of the adaptive linear model (2). It is well-known that when $\theta^* \in (-1, 1)$, the time series model (24) satisfies a stability assumption (1). Consequently, one might use the OLS estimate based confidence intervals [19] for $\theta^*$. However, when $\theta^* = 1$ — also known as the *unit root* case — stability condition (1) does not hold, and the

least squares estimator is not asymptotically normal [19]. In other words, when $\theta^* = 1$, the least squares based intervals do not provide correct coverage.

In this section, we apply ALEE-based approach to construct confidence intervals that are valid for $\theta^* \in [-1, 1]$. Similar to previous sections, let $s_0 \in \mathcal{F}_0$ and denote $s_t = s_0 + \sum_{1 \le i \le t} y_{i-1}^2$. Following a construction similar to the last section, we have the following corollary.

**Corollary 3.5.** *Assume the noise variables $\{\epsilon_t\}_t$ are i.i.d with mean zero, variance $\sigma^2$ and sub-Gaussian parameter $\sigma_g^2$. Then, for any $\theta^* \in [-1, 1]$, the ALEE estimator, obtained using $w_t = f(s_t/s_0)y_{t-1}/\sqrt{s_0}$ with function $f$ from Corollary 3.2 and $s_0 = n/\log\log(n)$, satisfies*

$$\sqrt{\frac{4\beta(\log 2)^\beta}{\log(s_n/s_0)\{\log\log(s_n/s_0)\}^{1+\beta}}} \cdot \frac{\sqrt{s_n}(\widehat{\theta}_{\text{ALEE}} - \theta^*)}{\widehat{\sigma}} \xrightarrow{d} \mathcal{N}(0, 1). \tag{25}$$

The proof of Corollary 3.5 can be found in Section A.3 of the Appendix.

## 3.3 Contextual bandits

In contextual bandit problems, the task of defining adaptive weights that satisfy the stability condition (6) while maintaining a large affinity is challenging. Without loss of generality, we assume that $\|\boldsymbol{x}_t\|_2 \le 1$. Following the discussion around (11) and using $\mathbf{S}_t$ as an approximation of $\mathbf{S}_n$, we see that a good choice for the weight is $\boldsymbol{w}_t \approx \mathbf{S}_t^{-\frac{1}{2}}\boldsymbol{x}_t$. However, it is not all clear at the moment why the above choice produces $d-$dimensional weights $\boldsymbol{w}_t$ satisfying the stability condition (6). It turns out that the success of our construction is based on the variability of certain matrix $\mathbf{V}_t$. For a $\mathcal{F}_0$-measurable $d \times d$ symmetric matrix $\boldsymbol{\Sigma}_0 \succeq \mathbf{I}_d$ and $t \in [n]$, we define

$$\boldsymbol{\Sigma}_t = \boldsymbol{\Sigma}_0 + \sum_{i=1}^{t} \boldsymbol{x}_i \boldsymbol{x}_i^\top \qquad \text{and} \qquad \boldsymbol{z}_t = \boldsymbol{\Sigma}_{t-1}^{-\frac{1}{2}} \boldsymbol{x}_t. \tag{26}$$

For $t \in [n]$, we define the variability matrix $\mathbf{V}_t$ as

$$\mathbf{V}_t = \left( \mathbf{I}_d + \sum_{i=1}^{t} \boldsymbol{z}_i \boldsymbol{z}_i^\top \right)^{-1} \qquad \text{(Variability)}. \tag{27}$$

The variability matrix $\mathbf{V}_t$ comes up frequently in finite sample analysis of the least squares estimator [16, 19, 2], the generalized linear models with adaptive data [23], and in online optimization [6]; see comments after Theorem 3.6 for a more detailed discussion on the matrix $\mathbf{V}_t$. Now, we define weights $\{\boldsymbol{w}_t\}_{t \ge 1}$ as

$$\boldsymbol{w}_t = \sqrt{1 + \boldsymbol{z}_t^\top \mathbf{V}_{t-1} \boldsymbol{z}_t} \cdot \mathbf{V}_t \boldsymbol{z}_t. \tag{28}$$

**Theorem 3.6.** *Suppose condition (3) holds and $\|\boldsymbol{\Sigma}_0^{-1}\|_{\text{op}} + \|\mathbf{V}_n\|_{\text{op}} = o_p(1)$. Then, the ALEE estimator $\widehat{\boldsymbol{\theta}}_{\text{ALEE}}$, obtained using the weights $\{\boldsymbol{w}_t\}_{1 \le t \le n}$ from (28), satisfies*

$$\left( \sum_{t=1}^{n} \boldsymbol{w}_t \boldsymbol{x}_t \right) \cdot (\widehat{\boldsymbol{\theta}}_{\text{ALEE}} - \boldsymbol{\theta}^*) \xrightarrow{d} \mathcal{N}(\mathbf{0}, \sigma^2 \mathbf{I}_d).$$

The proof of Theorem 3.6 can be found in Section A.4 of the Appendix. In Theorem B.4 in the appendix, we establish the asymptotic normality of a modified version of the ALEE estimator, which has the same asymptotic variance as the one in Theorem 3.6 under the assumption $\|\boldsymbol{\Sigma}_0^{-1}\|_{\text{op}} = o_p(1)$. In other words, the modified theorem B.4 does not assume any condition on the $\|\mathbf{V}_n\|_{\text{op}}$.

To better convey the idea of our construction, we provide a lemma that may be of independent interest. This lemma applies to weights $\boldsymbol{w}_t$ generated by (27) and (28) with general $\boldsymbol{z}_t$.

**Lemma 3.7.** *Let $\boldsymbol{w}_t$ be as in (28) with the variability matrix $\mathbf{V}_t$ in (27). Then,*

$$\sum_{t=1}^{n} \boldsymbol{w}_t \boldsymbol{w}_t^\top = \mathbf{I}_d - \mathbf{V}_n, \quad \max_{1 \le t \le n} \|\boldsymbol{w}_t\|_2 = \max_{1 \le t \le n} \|\mathbf{V}_{t-1} \boldsymbol{z}_t\|_2 / (1 + \boldsymbol{z}_t^\top \mathbf{V}_{t-1} \boldsymbol{z}_t)^{1/2}. \tag{29}$$

*For $\boldsymbol{z}_t \in \mathcal{F}_{t-1}$, the stability condition (6) holds when $\max_{1 \le t \le n} \boldsymbol{z}_t^\top \mathbf{V}_t \boldsymbol{z}_t + \|\mathbf{V}_n\|_{\text{op}} = o_p(1)$.*

**Proof.** For any $t \geq 1$, $\mathbf{V}_t = \mathbf{V}_{t-1} - \mathbf{V}_{t-1}\boldsymbol{z}_t\boldsymbol{z}_t^\top\mathbf{V}_{t-1}/(1 + \boldsymbol{z}_t^\top\mathbf{V}_{t-1}\boldsymbol{z}_t)$. It follows that $\mathbf{V}_t\boldsymbol{z}_t = \mathbf{V}_{t-1}\boldsymbol{z}_t/(1 + \boldsymbol{z}_t^\top\mathbf{V}_{t-1}\boldsymbol{z}_t)$ and $\sum_{t=1}^n \boldsymbol{w}_t\boldsymbol{w}_t^\top = \sum_{t=1}^n \mathbf{V}_{t-1}(\mathbf{V}_t^{-1} - \mathbf{V}_{t-1}^{-1})\mathbf{V}_t = \mathbf{I}_d - \mathbf{V}_n$.

**Comments on Theorem 3.6 conditions:** It is instructive to compare the conditions of Theorem 3.1 and Theorem 3.6. The condition $\|\boldsymbol{\Sigma}_0^{-1}\|_{\mathrm{op}} = o_p(1)$ is an analogue of the condition $1/s_0 = o_p(1)$. The condition $\|\mathbf{V}_n\|_{\mathrm{op}} = o_p(1)$ is a bit more subtle. This condition is an analogue of the condition $s_0/s_{n,k} = o_p(1)$. Indeed, applying elliptical potential lemma [2, Lemma 4] yields

$$\frac{\log(\det(\boldsymbol{\Sigma}_0 + \mathbf{S}_n))}{\log(\det(\boldsymbol{\Sigma}_0))} \leq \mathrm{trace}(\mathbf{V}_n^{-1}) - d = \sum_{t=1}^n \boldsymbol{x}_t^\top\boldsymbol{\Sigma}_{t-1}^{-1}\boldsymbol{x}_t \leq 2 \cdot \frac{\log(\det(\boldsymbol{\Sigma}_0 + \mathbf{S}_n))}{\log(\det(\boldsymbol{\Sigma}_0))} \tag{30}$$

where $\mathbf{S}_n = \sum_{i=1}^n \boldsymbol{x}_i\boldsymbol{x}_i^\top$ is the Gram matrix. We see that for $\|\mathbf{V}_n\|_{\mathrm{op}} = o_p(1)$, it is necessary that the eigenvalues of $\mathbf{S}_n$ grow to infinity at a faster rate than the eigenvalues of $\boldsymbol{\Sigma}_0$. Moreover, in the case of dimension $d = 1$, the condition $\|\mathbf{V}_n\|_{\mathrm{op}} = o_p(1)$ is equivalent to $s_0/s_{n,k} = o_p(1)$.

# 4 Numerical experiments

In this section, we consider three settings: two-armed bandit setting, first order auto-regressive model setting and contextual bandit setting. In two-armed bandit setting, the rewards are generated with same arm mean $(\theta_1^*, \theta_2^*) = (0.3, 0.3)$, and noise is generated from a normal distribution with mean $0$ and variance $1$. To collect two-armed bandit data, we use $\epsilon$-Greedy algorithm with decaying exploration rate $\sqrt{\log(t)/t}$. The rate is designed to make sure the number of times each armed is pulled has order greater than $\log(n)$ up to time $n$. In the second setting, we consider the time series model,

$$y_t = \theta^* y_{t-1} + \epsilon_t, \tag{31}$$

where $\theta^* = 1$ and noise $\epsilon_t$ is drawn from a normal distribution with mean $0$ and variance $1$. In the contextual bandit setting, we consider the true parameter $\boldsymbol{\theta}^*$ to be $0.3$ times the all-one vector. In the initial iterations, a random context $\boldsymbol{x}_t$ is generated from a uniform distribution in $\mathcal{S}^{d-1}$. Then, we apply $\epsilon$-Greedy algorithm to these pre-selected contexts with decaying exploration rate $\log^2(t)/t$. For all of the above three settings, we run $1000$ independent replications.

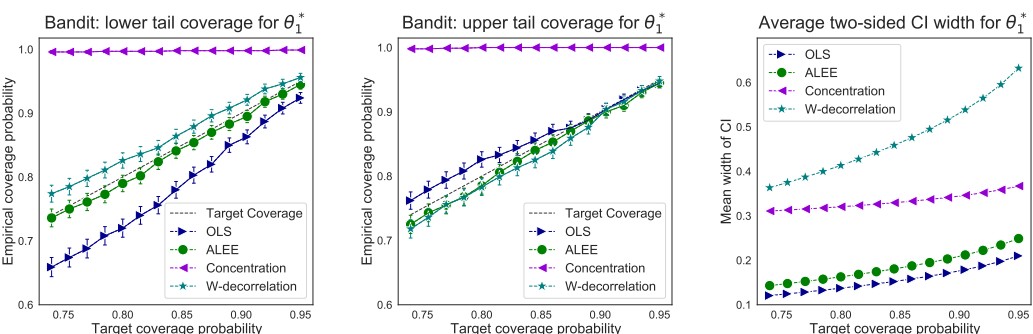

Figure 2: Two-armed bandit problem with equal arm mean $\theta_1^* = \theta_2^* = 0.3$. Error bars plotted are $\pm$ standard errors.

To analyze the data we collect for these settings, we apply ALEE approach with weights specified in Corollary 3.2, 3.5 and Theorem B.4, respectively. More specifically, in the first two settings, we consider $\beta = 1$ in Corollary 3.2. For two-armed bandit example, we set $s_0 = e^2\log(n)$, which is known to be a lower bound for $s_{n,1}$. For AR(1) model, we consider $s_0 = e^2 n/\log\log(n)$. For the contextual bandit example, we consider $\boldsymbol{\Sigma}_0 = \log(n) \cdot \mathbf{I}_d$. In the simulations, we also compare ALEE approach to the normality based confidence interval for OLS estimator [19] (which may be incorrect), the concentration bounds for the OLS estimator based on self-normalized martingale sequence [1], and W-decorrelation [8]. Detailed implementations about these methods can be found in Appendix C.1.

In Figure 2, we display results for two-armed bandit example, providing the empirical coverage plots for the first arm mean $\theta_1^*$ as well as average width for two-sided CIs. We observe that CIs based on

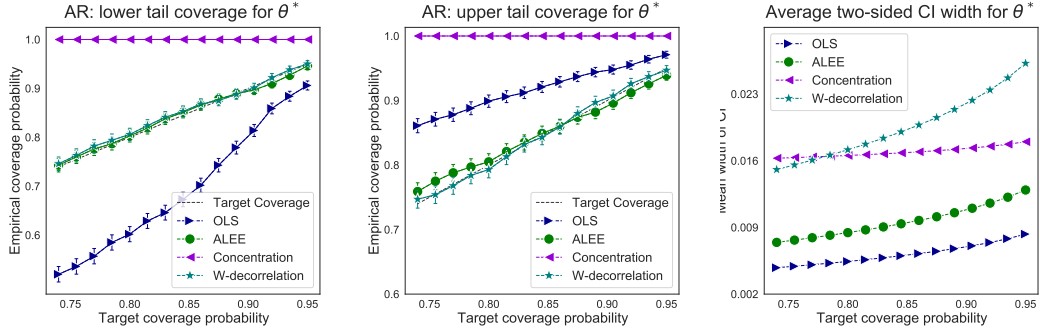

Figure 3: AR(1) with model coefficient $\theta^* = 1$ and $s_0 = e^2 n / \log \log(n)$. Error bars plotted are $\pm$ standard errors.

OLS undercover $\theta_1^*$ while other methods provide satisfactory coverage. Notably, from the average CI width plot, we can see that W-decorrelation and concentration methods have relatively large CI widths. On the contrary, ALEE-based CIs achieve target coverage while keeping the width of CIs relatively small.

For AR(1) model, we display the results in Figure 3. For the context bandit example, we consider $d = 20$ and summarize the empirical coverage probability and the logarithm of the volume of the confidence regions in Table 1, along with corresponding standard deviations. See Appendix C.2 for experiments with dimension $d = 10$ and $d = 50$.

Table 1: Contextual bandit: d = 20

| Method | Level of confidence | | | | | |
|---|---|---|---|---|---|---|
| | 0.8 | | 0.85 | | 0.9 | |
| | Avg. Coverage | Avg. log(Volumn) | Avg. Coverage | Avg. log(Volumn) | Avg. Coverage | Avg. log(Volumn) |
| ALEE | 0.805 ($\pm$ 0.396) | 6.541 ($\pm$ 0.528) | 0.861 ($\pm$ 0.346) | 7.108 ($\pm$ 0.528) | 0.910 ($\pm$ 0.286) | 7.806 ($\pm$ 0.528) |
| OLS | 0.776 ($\pm$ 0.417) | -2.079 ($\pm$ 0.525) | 0.830 ($\pm$ 0.376) | -1.513 ($\pm$ 0.525) | 0.881 ($\pm$ 0.324) | -0.815 ($\pm$ 0.525) |
| W-Decorrelation | 0.777 ($\pm$ 0.416) | 25.727 ($\pm$ 0.518) | 0.829 ($\pm$ 0.377) | 26.294 ($\pm$ 0.518) | 0.870 ($\pm$ 0.336) | 26.992 ($\pm$ 0.518) |
| Concentration | 1.000 ($\pm$ 0.000) | 17.374 ($\pm$ 0.506) | 1.000 ($\pm$ 0.000) | 17.408($\pm$ 0.506) | 1.000 ($\pm$ 0.000) | 17.455 ($\pm$ 0.506) |

## 5 Discussion

In this paper, we study the parameter estimation problem in an adaptive linear model. We propose to use ALEE (adaptive linear estimation equations) to obtain point and interval estimates. Our main contribution is to propose an estimator which is asymptotically normal without requiring any stability condition on the sample covariance matrix. Unlike the concentration based confidence regions, our proposed confidence regions allow for heavy tailed noise variables. We demonstrate the utility of our method by comparing our method with existing methods.

Our work leaves several questions open for future research. For example, it would be interesting to characterize the variance of the ALEE estimator compared to the best possible variance[13, 20] for $d > 1$. It would also be interesting to know if such results can be extended to non-linear adaptive models, e.g., to an adaptive generalized linear model [23]. Furthermore, our paper assumes a fixed dimension $d$ for the problem while letting $n \to \infty$. It would be interesting to explore whether we can allow the dimension to grow with the number of samples at a specific rate.

## Acknowledgments

This work was partially supported by the National Science Foundation Grants DMS-2311304, CCF-1934924, DMS-2052949 and DMS-2210850.

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

# Appendix

## Table of Contents

## A  Proof

In Theorem 3.1, Corollary 3.3 and Remark 3.4, we deal with an arm with index $k \in [K]$. To simplify notations, we drop the subscript $k$ in $s_{t,k}, w_{t,k}, x_{t,k}, \widehat{\theta}_{\mathrm{ALEE},k}$ and $\theta_k^*$ throughout the proof, and use $s_t, w_t, x_t, \widehat{\theta}_{\mathrm{ALEE}}$ and $\theta^*$, respectively.

### A.1  Proof of Theorem 3.1

Condition (17) serves as an important role in proving (18). Therefore, we start our proof by verifying the condition (17). Since function $f$ is a positive decreasing function, we first have

$$\max_{1 \le t \le n} f^2(\frac{s_t}{s_0})\frac{x_t^2}{s_0} \le f^2(1)\frac{1}{s_0}. \tag{32}$$

Furthermore, since function $f'/f$ is increasing, we have

$$\begin{aligned}
\max_{1 \le t \le n} \left( 1 - \frac{f(s_t/s_0)}{f(s_{t-1}/s_0)} \right) &= \max_{1 \le t \le n} \frac{f(s_{t-1}/s_0) - f(s_t/s_0)}{f(s_{t-1}/s_0)} \\
&\overset{(i)}{\le} \frac{1}{s_0} \max_{1 \le t \le n} \frac{-f'(s_{t-1}/s_0)}{f(s_{t-1}/s_0)} = \frac{1}{s_0} \frac{-f'(1)}{f(1)},
\end{aligned} \tag{33}$$

where inequality $(i)$ follows from mean value theorem and the monotonicity of the function $f'/f$. Thus, by assuming $1/s_0 = o_p(1)$ and $s_0/s_n = o_p(1)$, condition (17) follows directly from equation (32) and equation (33).

By the construction of ALEE estimator, we have

$$\left\{ \sum_{t=1}^n w_t x_t \right\} \cdot (\widehat{\theta}_{\mathrm{ALEE}} - \theta^*) = \sum_{t=1}^n w_t \epsilon_t. \tag{34}$$

Note that

$$\sum_{t=1}^n w_t x_t = \sqrt{s_0} \sum_{t=1}^n f(s_t/s_0)\frac{x_t^2}{s_0} = \sqrt{s_0} \int_1^{s_n/s_0} f(x) dx \cdot \frac{\sum_{t \le n} f(s_t/s_0)x_t^2/s_0}{\int_1^{s_n/s_0} f(x) dx}. \tag{35}$$

By the mean value theorem, we have that for $t \in [n]$, $\xi_t \in [s_{t-1}, s_t]$

$$\int_{s_{t-1}/s_0}^{s_t/s_0} f(x) dx = f(\xi_t/s_0)\frac{x_t^2}{s_0}.$$

Therefore, we have

$$\frac{\sum_{t \leq n} f(s_t/s_0) x_t^2/s_0}{\int_1^{s_n/s_0} f(x) dx} = 1 + \underbrace{\frac{\sum_{t \leq n} (\frac{f(s_t/s_0)}{f(\xi_t/s_0)} - 1) f(\xi_t/s_0) x_t^2/s_0}{\sum_{t \leq n} f(\xi_t/s_0) x_t^2/s_0}}_{\triangleq R}.$$

Observe that

$$|R| \leq \frac{\sum_{t \leq n} |\frac{f(s_t/s_0)}{f(\xi_t/s_0)} - 1| f(\xi_t/s_0) x_t^2/s_0}{\sum_{t \leq n} f(\xi_t/s_0) x_t^2/s_0}$$

$$\leq \frac{\sum_{t \leq n} |\frac{f(s_t/s_0)}{f(s_{t-1}/s_0)} - 1| f(\xi_t/s_0) x_t^2/s_0}{\sum_{t \leq n} f(\xi_t/s_0) x_t^2/s_0}$$

$$\leq \max_{1 \leq t \leq n} \left(1 - \frac{f(s_t/s_0)}{f(s_{t-1}/s_0)}\right) \overset{(ii)}{=} o_p(1).$$

Equality $(ii)$ follows from condition (17). Consequently, applying Slutsky's theorem yields

$$\frac{\sum_{i=1}^n w_t x_t}{\sqrt{s_0} \int_1^{s_n/s_0} f(x) dx} \xrightarrow{p} 1.$$

Similarly, we can derive

$$\sum_{t=1}^n w_t^2 = \sum_{t=1}^n f^2(s_t/s_0) \frac{x_t^2}{s_0} = (1 + o_p(1)) \int_1^{s_n/s_0} f^2(x) dx = 1 + o_p(1). \tag{36}$$

Knowing $\max_{1 \leq t \leq n} w_t^2 = \max_{1 \leq t \leq n} f^2(s_t/s_0) x_t^2/s_0 = o_p(1)$, which is a consequence of equation (17), martingale central limit theorem together with an application of Slutsky's theorem yields

$$(\widehat{\theta}_{\text{ALEE}} - \theta^*) \cdot \int_1^{s_n/s_0} \frac{\sqrt{s_0}}{\widehat{\sigma}} f(x) dx \xrightarrow{d} \mathcal{N}(0, 1).$$

Lastly, we recall that

$$\frac{\widehat{\theta}_{\text{ALEE}} - \theta^*}{\widehat{\sigma} \sqrt{\sum_{t \leq n} w_t^2}} \cdot \left(\sum_{t=1}^n w_t x_t\right) = \frac{1}{\widehat{\sigma} \sqrt{\sum_{t \leq n} w_t^2}} \sum_{t=1}^n w_t \epsilon_t.$$

Therefore, equation (19) follows from martingale central limit theorem and Slutsky's theorem.

**Remark A.1.** *Equation (18) sheds light on the asymptotic variance of the ALEE estimator, thereby aiding in the selection of a suitable function f to improve the efficiency of ALEE estimator. On the other hand, equation (19) offers a practical approach to obtaining an asymptotically precise confidence interval.*

**Remark A.2.** *Condition (17) is a general requirement that governs equation (18), and is not specific to bandit problems. However, the difficulty in verifying (17) can vary depending on the problem at hand.*

### A.2   Proof of Remark 3.4

Corollary 3.3 follows directly from Theorem 1 in [1]. In this section, we provide a proof of Remark 3.4. By considering $\lambda_0 = 1$ in Corollary 3.3, we have with probability at least $1 - \delta$

$$\left|\sum_{t=1}^n w_t x_t\right| \cdot |\widehat{\theta}_{\text{ALEE}} - \theta^*| \leq \sigma_g \sqrt{\left(1 + \sum_{t=1}^n w_t^2\right) \cdot \log\left(\frac{1 + \sum_{t=1}^n w_t^2}{\delta^2}\right)}. \tag{37}$$

By the construction of the weights in Corollary 3.2, we have

$$\sum_{t=1}^n w_t^2 = \sum_{t=1}^n f^2(\frac{s_t}{s_0}) \frac{x_t^2}{s_0} \leq \int_1^\infty f^2(x) dx = 1. \tag{38}$$

Therefore, to complete the proof, it suffices to characterize a lower bound for $\sum_{1 \le t \le n} w_t x_t$. By definition, we have

$$
\begin{aligned}
\sum_{t=1}^{n} w_t x_t &= \sum_{t=1}^{n} f(s_t/s_0) \frac{x_t^2}{\sqrt{s_0}} \\
&\overset{(i)}{=} \sum_{t=1}^{n} \frac{x_t^2}{(s_t \log(e^2 s_t/s_0))^{1/2} \log\log(e^2 s_t/s_0)} \\
&\ge \frac{1}{(2 + \log(s_n/s_0))^{1/2} \log(2 + \log(s_n/s_0))} \sum_{t=1}^{n} \frac{x_t^2}{\sqrt{s_t}} \\
&\overset{(ii)}{\ge} \frac{1}{(2 + \log(s_n/s_0))^{1/2} \log(2 + \log(s_n/s_0))} \cdot 2(\sqrt{s_n} - \sqrt{s_0}) \sqrt{\frac{s_0}{1 + s_0}} \\
&\overset{(iii)}{\ge} \frac{1}{(2 + \log(s_n/s_0))^{1/2} \log(2 + \log(s_n/s_0))} \cdot \sqrt{2}(\sqrt{s_n} - \sqrt{s_0}).
\end{aligned}
\tag{39}
$$

In equation $(i)$, we plug in the expression of function $f$ and hence $\sqrt{s_0}$ cancels out. Since $x_t$ is either 0 or 1, inequality $(ii)$ follows from the integration of the function $h(x) = 1/\sqrt{x}$. Inequality $(iii)$ follows from $s_0 > 1$. Putting things together, we have

$$
\begin{aligned}
|\widehat{\theta}_{\mathrm{ALEE}} - \theta^*| &\le \sigma_g \frac{\sqrt{2 \log(2/\delta^2)}}{\sum_{1 \le t \le n} w_t x_t} \\
&\le \sigma_g \sqrt{\log(2/\delta^2)} \frac{\sqrt{2 + \log(s_n/s_0)} \log\{2 + \log(s_n/s_0)\}}{\sqrt{s_n} - \sqrt{s_0}}.
\end{aligned}
\tag{40}
$$

This completes our proof of Remark 3.4.

## A.3 Proof of Corollary 3.5

Note that it suffices to verify the following condition (41)

$$
\max_{1 \le t \le n} f^2\left(\frac{s_t}{s_0}\right) \frac{y_{t-1}^2}{s_0} + \max_{1 \le t \le n} \left(1 - \frac{f(s_t/s_0)}{f(s_{t-1}/s_0)}\right) + \int_{s_n/s_0}^{\infty} f^2(x) dx = o_p(1)
\tag{41}
$$

for $\theta^* \in [-1, 1]$ in order to complete the proof of Corollary 3.5. The other part of the proof can be adapted from the proof of Theorem 3.1. To simplify notations, we let

$$
T_1 \overset{\triangle}{=} \max_{1 \le t \le n} f^2\left(\frac{s_t}{s_0}\right) \frac{y_{t-1}^2}{s_0}, \quad T_2 \overset{\triangle}{=} \max_{1 \le t \le n} \left(1 - \frac{f(s_t/s_0)}{f(s_{t-1}/s_0)}\right), \quad \text{and} \quad T_3 \overset{\triangle}{=} \int_{s_n/s_0}^{\infty} f^2(x) dx.
$$

Therefore, proving equation (41) is equivalent to showing that $T_1$, $T_2$, and $T_3$ converge to zero in probability. We will now demonstrate the convergence of each of these three terms to zero in probability.

$T_1$ **with** $\theta^* = 1$: To prove $T_1 = o_p(1)$, we make use of a result in [19, Equation 3.23], which is

$$
\mathbb{P}\left( \liminf_{n \to \infty} n^{-2} (\log\log n) \sum_{t=1}^{n} y_{t-1}^2 = \sigma^2/4 \right) = 1.
\tag{42}
$$

Observe that

$$
\begin{aligned}
T_1 = \max_{1 \le t \le n} f^2\left(\frac{s_t}{s_0}\right) \frac{y_{t-1}^2}{s_0} &= \max_{1 \le t \le n} \frac{y_{t-1}^2}{s_t \log(e^2 s_t/s_0)\{\log\log(e^2 s_t/s_0)\}^{1+\beta}} \\
&\le \max_{1 \le t \le n} \frac{y_{t-1}^2}{s_t \log(e^2)\{\log\log(e^2)\}^{1+\beta}} \\
&= \frac{1}{2(\log 2)^{1+\beta}} \max_{1 \le t \le n} \frac{y_{t-1}^2}{s_t} \\
&\le \frac{1}{2(\log 2)^{1+\beta}} \max\{ \max_{1 \le t \le \lfloor n^{2/3} \rfloor} \frac{y_{t-1}^2}{s_0}, \max_{\lfloor n^{2/3} \rfloor + 1 \le t \le n} \frac{y_{t-1}^2}{s_{\lfloor n^{2/3} \rfloor} - s_0} \}
\end{aligned}
\tag{43}
$$

In equation (43), we split the sequence into two parts and set different lower bounds for $s_t$. The major benefit of this step is to help us derive a better choice of $s_0$. Now we bound $\max_{1 \le t \le \lfloor n^{2/3} \rfloor} y_{t-1}^2$ and

$\max_{\lfloor n^{2/3} \rfloor + 1 \leq t \leq n} y_{t-1}^2$. Note that

$$\mathbb{P}\left(\max_{1 \leq t \leq \lfloor n^{2/3} \rfloor} y_{t-1}^2 \geq \epsilon\right)$$

$$=\mathbb{P}\left(\max\{\max_{1 \leq t \leq \lfloor n^{2/3} \rfloor} y_{t-1}, \max_{1 \leq t \leq \lfloor n^{2/3} \rfloor} -y_{t-1}\} \geq \sqrt{\epsilon}\right)$$

$$\leq \frac{\mathbb{E}\left[\max\{\max_{1 \leq t \leq \lfloor n^{2/3} \rfloor} y_{t-1}, \max_{1 \leq t \leq \lfloor n^{2/3} \rfloor} -y_{t-1}\}\right]}{\sqrt{\epsilon}}$$

$$\overset{(i)}{\leq} \sqrt{\frac{2n^{2/3} \sigma_g^2 \log(2n^{2/3})}{\epsilon}}, \tag{44}$$

where inequality is derived from [33, Exercise 2.12] and the fact that $y_i$ is sub-Gaussian with sub-Gaussian parameter $\sigma_g^2 n^{2/3}$ for $i \leq \lfloor n^{2/3} \rfloor$. Therefore, we conclude that

$$\max_{1 \leq t \leq \lfloor n^{2/3} \rfloor} y_{t-1}^2 = O_p(n^{2/3} \log n).$$

Consequently, we have

$$\max_{1 \leq t \leq \lfloor n^{2/3} \rfloor} \frac{y_{t-1}^2}{s_0} = \frac{n^{2/3} \log n}{n / \log \log n} \cdot O_p(1) = o_p(1). \tag{45}$$

By applying the same trick to $\max_{\lfloor n^{2/3} \rfloor + 1 \leq t \leq n} y_{t-1}^2$, we can derive

$$\max_{\lfloor n^{2/3} \rfloor + 1 \leq t \leq n} y_{t-1}^2 = O_p(n \log n).$$

Hence we have

$$\max_{\lfloor n^{2/3} \rfloor + 1 \leq t \leq n} \frac{y_{t-1}^2}{s_{\lfloor n^{2/3} \rfloor} - s_0} = \frac{O_p(n \log n)}{n^{4/3} / \log \log n^{2/3}} \cdot \frac{n^{4/3} / \log \log n^{2/3}}{s_{\lfloor n^{2/3} \rfloor} - s_0} \overset{(ii)}{=} o_p(1) \cdot O_p(1) = o_p(1). \tag{46}$$

Equality $(ii)$ makes use of equation (42). Combining equation (44) with equations (45) and (46), we conclude that $T_1 = o_p(1)$.

$T_1$ **with $\theta^* = -1$:** When $\theta^* = -1$, the proof is essentially the same as the case when $\theta^* = -1$. The only difference lies in the order of $\sum_{1 \leq i \leq n} y_{i-1}^2$. However, by pairing $\epsilon_{2t-1}$ with $\epsilon_{2t}$ for $t \in \mathbb{N}^+$, we can arrive at the same result. Specifically, for $t \in \mathbb{N}^+$, we let $\epsilon_t' = \epsilon_{2t} - \epsilon_{2t-1}$ and define

$$y_t' = \sum_{k=1}^{t} \epsilon_k'$$

where $y_0' \triangleq 0$ and $\{\epsilon_t'\}_{t \geq 1}$ are random variables with mean zero, variance $2\sigma^2$ and sub-Gaussian parameter $2\sigma_g^2$. Therefore, applying equation (44) yields

$$\liminf_{n \to \infty} n^{-2}(\log \log n) \sum_{t=1}^{n} (y_{t-1}')^2 = \sigma^2. \tag{47}$$

Setting $n_0 = \lfloor (\lfloor n^{2/3} \rfloor - 1)/2 \rfloor$, we have

$$s_{\lfloor n^{2/3} \rfloor} - s_0 = \sum_{t=1}^{\lfloor n^{2/3} \rfloor - 1} y_t^2 \geq \sum_{t=1}^{n_0} (y_t')^2 = \sum_{t=1}^{n_0+1} (y_{t-1}')^2. \tag{48}$$

According to equation (47) and equation (48), we have

$$\max_{\lfloor n^{2/3} \rfloor + 1 \leq t \leq n} \frac{y_{t-1}^2}{s_{\lfloor n^{2/3} \rfloor} - s_0} \leq \frac{\max_{\lfloor n^{2/3} \rfloor + 1 \leq t \leq n} y_{t-1}^2}{\sum_{1 \leq t \leq n_0+1} (y_{t-1}')^2}$$

$$= \frac{\max_{\lfloor n^{2/3} \rfloor + 1 \leq t \leq n} y_{t-1}^2}{(n_0 + 1)^2 / \log \log(n_0 + 1)} \cdot \frac{(n_0 + 1)^2 / \log \log(n_0 + 1)}{\sum_{1 \leq t \leq n_0+1} (y_{t-1}')^2} \tag{49}$$

$$= o_p(1) \cdot O_p(1) = o_p(1),$$

which completes the proof of $T_1 = o_p(1)$ for the case when $\theta^* = -1$.

$T_1$ **with** $\theta^* \in (-1, 1)$**:** Given $\theta^* \in (-1, 1)$, we observe that $y_t$ is a sub-Gaussian random variable with sub-Gaussian parameter $\frac{\sigma_g^2}{1-(\theta^*)^2}$ for any $t \in \mathbb{N}^+$. Therefore, following equation (43), we have

$$T_1 \le \frac{1}{2(\log 2)^{1+\beta}} \max_{1 \le t \le n} \frac{y_{t-1}^2}{s_0} \tag{50}$$

where in the above inequality we use $s_0$ as a lower bound for $s_t$. By applying [33, Exercise 2.12], we have

$$\max_{1 \le t \le n} y_{t-1}^2 = O_p(\log n), \tag{51}$$

leading to the conclusion that $T_1 = o_p(1)$.

$T_2$ **with** $\theta^* \in [-1, 1]$**:** Similar to equation (33), we have

$$T_2 = \max_{1 \le t \le n} \left( 1 - \frac{f(s_t/s_0)}{f(s_{t-1}/s_0)} \right) = \max_{1 \le t \le n} \frac{f(s_{t-1}/s_0) - f(s_t/s_0)}{f(s_{t-1}/s_0)}$$

$$\le \max_{1 \le t \le n} \frac{-f'(s_{t-1}/s_0)}{f(s_{t-1}/s_0)} \frac{y_{t-1}^2}{s_0}.$$

Define $g(x) = -f'(x)/f(x)$ and we can compute that

$$\int g(x)dx = -\int \frac{f'(x)}{f(x)}dx = -\int \frac{1}{f}df = -\log f + C,$$

where $C$ is some constant. Doing some calculation yields

$$g(x) = \frac{d}{dx} - \log f = \frac{d}{dx} \left\{ \frac{1}{2} (\log(x) + \log \log(e^2 x)) + (1 + \beta) \log \log \log(e^2 x) \right\}$$

$$= \frac{1}{2x} \left\{ 1 + \frac{1}{\log(e^2 x)} + \frac{1+\beta}{\log(e^2 x)} \cdot \frac{1}{\log \log(e^2 x)} \right\}.$$

Therefore, we have

$$T_2 \le \max_{1 \le t \le n} \frac{-f'(s_{t-1}/s_0)}{f(s_{t-1}/s_0)} \frac{y_{t-1}^2}{s_0} = \max_{1 \le t \le n} g(s_{t-1}/s_0) \frac{y_{t-1}^2}{s_0}$$

$$\le \frac{1}{2} \left( \frac{3}{2} + \frac{1+\beta}{2\log 2} \right) \max_{1 \le t \le n} \frac{y_{t-1}^2}{s_{t-1}}.$$

We note that demonstrating $\max_{1 \le t \le n} y_{t-1}^2/s_{t-1} = o_p(1)$ follows the same approach as the proof of $\max_{1 \le t \le n} y_{t-1}^2/s_t = o_p(1)$. Hence, we omit it. To conclude, we show that $T_2 = o_p(1)$ for $\theta^* \in [-1, 1]$.

$T_3$ **with** $\theta^* \in [-1, 1]$**:** To prove $T_3 = o_p(1)$, it suffices to verify that

$$\frac{s_0}{\sum_{1 \le t \le n} y_t^2} = o_p(1). \tag{52}$$

For convenience, in equation (52) we use $y_t$ instead of $y_{t-1}$. Note that when $\theta^* = 1$ or $\theta^* = -1$, we have provided almost sure lower bounds for $\sum_{1 \le t \le n} y_t^2$ in the proof of $T_1 = o_p(1)$. Therefore, equation (52) follows from these lower bounds. To prove equation (52) when $\theta^* \in (-1, 1)$, we begin by rewriting $\sum_{1 \le t \le n} y_t^2$ in quadratic form. Without confusion and loss of generality, we replace $\theta^*$ by $\theta$, consider $\text{Var}(\epsilon_t) = 1$, and set $\boldsymbol{\varepsilon}_n = (\epsilon_1, \epsilon_2, \ldots, \epsilon_n)^\top$. For $t \in [n]$, we have

$$y_t = \sum_{k=1}^t \theta^{t-k} \epsilon_k = \boldsymbol{a}_t^\top \boldsymbol{\varepsilon}_n,$$

where $\boldsymbol{a}_t \in \mathbb{R}^n$ and $a_{t,j} = \theta^{t-j}$ for $j \le t$ and $a_{t,j} = 0$ for $j > t$. Therefore, $\sum_{1 \le t \le n} y_t^2$ can be written as

$$\sum_{1 \le t \le n} y_t^2 = \sum_{1 \le t \le n} \boldsymbol{\varepsilon}_n^\top \boldsymbol{a}_t \boldsymbol{a}_t^\top \boldsymbol{\varepsilon}_n = \boldsymbol{\varepsilon}_n^\top \mathbf{A} \boldsymbol{\varepsilon}_n, \tag{53}$$

where $\mathbf{A} = \sum_{1 \le t \le n} \boldsymbol{a}_t \boldsymbol{a}_t^\top$. Applying Hanson-Wright inequality (e.g. see [32]), we have

$$\mathbb{P} \left( |\boldsymbol{\varepsilon}_n^\top \mathbf{A} \boldsymbol{\varepsilon}_n - \mathbb{E} \boldsymbol{\varepsilon}_n^\top \mathbf{A} \boldsymbol{\varepsilon}_n| > t \right) \le 2 \exp \left[ -c \min \left( \frac{t^2}{K^4 \|\mathbf{A}\|_F^2}, \frac{t}{K^2 \|\mathbf{A}\|_F} \right) \right], \tag{54}$$

where $c$ and $K$ are some universal constants. Observe that

$$\mathbb{E}\boldsymbol{\varepsilon}_n^\top \mathbf{A} \boldsymbol{\varepsilon}_n = \text{trace}(A) = \text{trace}(\sum_{1 \le t \le n} \boldsymbol{a}_t \boldsymbol{a}_t^\top) = \text{trace}(\sum_{1 \le t \le n} \boldsymbol{a}_t^\top \boldsymbol{a}_t)$$

$$= \sum_{1 \le t \le n} (1 + \theta^2 + \cdots + \theta^{2(t-1)})$$

$$= \sum_{1 \le t \le n} \frac{1 - \theta^{2t}}{1 - \theta^2}$$

$$= \frac{n}{1 - \theta^2} - \frac{\theta^2(1 - \theta^{2n})}{(1 - \theta^2)^2}.$$

Furthermore, we have

$$\|\mathbf{A}\|_F^2 = \text{trace}(\mathbf{A}^\top \mathbf{A}) = \text{trace}(\sum_{1 \le i \le n} \boldsymbol{a}_i \boldsymbol{a}_i^\top \cdot \sum_{1 \le j \le n} \boldsymbol{a}_j \boldsymbol{a}_j^\top)$$

$$= \sum_{1 \le i \le n} \sum_{1 \le j \le n} (\boldsymbol{a}_i^\top \boldsymbol{a}_j)^2 \tag{55}$$

$$= \sum_{1 \le i \le n} \|\boldsymbol{a}_i\|_2^4 + 2 \sum_{1 \le i < j \le n} \|\boldsymbol{a}_i\|_2^4 \cdot \theta^{2(j-i)}.$$

Subsequently, we have

$$\sum_{1 \le i \le n} \|\boldsymbol{a}_i\|_2^4 \le \|\mathbf{A}\|_F^2 \le (1 + \frac{2}{1 - \theta^2}) \sum_{1 \le i \le n} \|\boldsymbol{a}_i\|_2^4, \tag{56}$$

where

$$\sum_{1 \le i \le n} \|\boldsymbol{a}_i\|_2^4 = \frac{n}{(1 - \theta^2)^2} - \frac{2\theta^2(1 - \theta^{2n})}{(1 - \theta^2)^3} + \frac{\theta^4(1 - \theta^{4n})}{(1 - \theta^2)^2(1 - \theta^4)}.$$

Assuming $\delta \le 2e^{-c}$ and $t = \frac{1}{c} K^2 \|\mathbf{A}\|_F \log(\frac{2}{\delta})$, we have with probability at least $1 - \delta$,

$$\boldsymbol{\varepsilon}_n^\top \mathbf{A} \boldsymbol{\varepsilon}_n \ge \mathbb{E}\boldsymbol{\varepsilon}_n^\top \mathbf{A} \boldsymbol{\varepsilon}_n - \frac{1}{c} K^2 \|\mathbf{A}\|_F \log(\frac{2}{\delta}). \tag{57}$$

We note that the term on the right hand side of equation (57) has order $n$. For any $\epsilon > 0$, consider the following probability

$$\limsup_{n \to \infty} \mathbb{P}\left(\frac{s_0}{\boldsymbol{\varepsilon}_n^\top \mathbf{A} \boldsymbol{\varepsilon}_n} > \epsilon\right) \le \limsup_{n \to \infty} \mathbb{P}\left(\frac{s_0}{\boldsymbol{\varepsilon}_n^\top \mathbf{A} \boldsymbol{\varepsilon}_n} > \epsilon, \boldsymbol{\varepsilon}_n^\top \mathbf{A} \boldsymbol{\varepsilon}_n \ge \mathbb{E}\boldsymbol{\varepsilon}_n^\top \mathbf{A} \boldsymbol{\varepsilon}_n - \frac{1}{c} K^2 \|\mathbf{A}\|_F \log(\frac{2}{\delta})\right)$$

$$+ \limsup_{n \to \infty} \mathbb{P}\left(\boldsymbol{\varepsilon}_n^\top \mathbf{A} \boldsymbol{\varepsilon}_n < \mathbb{E}\boldsymbol{\varepsilon}_n^\top \mathbf{A} \boldsymbol{\varepsilon}_n - \frac{1}{c} K^2 \|\mathbf{A}\|_F \log(\frac{2}{\delta})\right) \tag{58}$$

$$\le \limsup_{n \to \infty} \mathbb{P}\left(\frac{s_0}{\mathbb{E}\boldsymbol{\varepsilon}_n^\top \mathbf{A} \boldsymbol{\varepsilon}_n - \frac{1}{c} K^2 \|\mathbf{A}\|_F \log(\frac{2}{\delta})} > \epsilon\right) + \delta.$$

By fixing $\delta$ and comparing the order of $s_0$ with the order of $\boldsymbol{\varepsilon}_n^\top \mathbf{A} \boldsymbol{\varepsilon}_n - \frac{1}{c} K^2 \|\mathbf{A}\|_F \log(\frac{2}{\delta})$, we have

$$\limsup_{n \to \infty} \mathbb{P}\left(\frac{s_0}{\mathbb{E}\boldsymbol{\varepsilon}_n^\top \mathbf{A} \boldsymbol{\varepsilon}_n - \frac{1}{c} K^2 \|\mathbf{A}\|_F \log(\frac{2}{\delta})} > \epsilon\right) = 0.$$

Since $\delta$ can be arbitrarily small, we conclude that

$$\frac{s_0}{\boldsymbol{\varepsilon}_n^\top \mathbf{A} \boldsymbol{\varepsilon}_n} = o_p(1), \tag{59}$$

which completes the proof of $T_3 = o_p(1)$.

### A.4 Proof of Theorem 3.6

Note that for any $t \ge 1$, we have

$$\|\mathbf{V}_t\|_{\text{op}} \le 1 \quad \text{and} \quad \mathbf{V}_t = \mathbf{V}_{t-1} - \mathbf{V}_{t-1} \boldsymbol{z}_t \boldsymbol{z}_t^\top \mathbf{V}_{t-1} / (1 + \boldsymbol{z}_t^\top \mathbf{V}_{t-1} \boldsymbol{z}_t). \tag{60}$$

The second part of equation (60) follows from the Sherman–Morrison formula. Let $\boldsymbol{u}_t = \mathbf{V}_t \boldsymbol{z}_t$ and we adopt the notation $\mathbf{V}_0 = \mathbf{I}_d$. By multiplying $\boldsymbol{z}_t$ on the right hand side of $\mathbf{V}_t$, we have

$$\mathbf{V}_t \boldsymbol{z}_t = \mathbf{V}_{t-1} \boldsymbol{z}_t - \mathbf{V}_{t-1} \boldsymbol{z}_t \boldsymbol{z}_t^\top \mathbf{V}_{t-1} \boldsymbol{z}_t / (1 + \boldsymbol{z}_t^\top \mathbf{V}_{t-1} \boldsymbol{z}_t)$$

$$= \mathbf{V}_{t-1} \boldsymbol{z}_t \left(1 - \frac{\boldsymbol{z}_t^\top \mathbf{V}_{t-1} \boldsymbol{z}_t}{1 + \boldsymbol{z}_t^\top \mathbf{V}_{t-1} \boldsymbol{z}_t}\right) = \frac{\mathbf{V}_{t-1} \boldsymbol{z}_t}{1 + \boldsymbol{z}_t^\top \mathbf{V}_{t-1} \boldsymbol{z}_t}. \tag{61}$$

Therefore, following the definition of $\boldsymbol{u}_t$, we have $(1 + \boldsymbol{z}_t^\top \mathbf{V}_{t-1} \boldsymbol{z}_t) \boldsymbol{u}_t = \mathbf{V}_{t-1} \boldsymbol{z}_t$. Consequently,

$$\sum_{t=1}^{n} (1 + \boldsymbol{z}_t^\top \mathbf{V}_{t-1} \boldsymbol{z}_t) \boldsymbol{u}_t \boldsymbol{u}_t^\top = \sum_{t=1}^{n} \mathbf{V}_{t-1} (\mathbf{V}_t^{-1} - \mathbf{V}_{t-1}^{-1}) \mathbf{V}_t = \mathbf{I}_d - \mathbf{V}_n. \tag{62}$$

By recognizing $\boldsymbol{w}_t = \sqrt{1 + \boldsymbol{z}_t^\top \mathbf{V}_{t-1} \boldsymbol{z}_t} \cdot \boldsymbol{u}_t$, we come to

$$\sum_{t=1}^{n} \boldsymbol{w}_t \boldsymbol{w}_t^\top = \sum_{t=1}^{n} \mathbf{V}_{t-1} (\mathbf{V}_t^{-1} - \mathbf{V}_{t-1}^{-1}) \mathbf{V}_t = \mathbf{I}_d - \mathbf{V}_n.$$

What remains now is to verify conditions in (6). Notably, assumption $\|\mathbf{V}_n\|_{\mathrm{op}} = o_p(1)$ implies

$$\sum_{t=1}^{n} \boldsymbol{w}_t \boldsymbol{w}_t^\top \xrightarrow{p} \mathbf{I}_d. \tag{63}$$

Since $\|\boldsymbol{\Sigma}_0^{-1}\|_{\mathrm{op}} = o_p(1)$, $\|\mathbf{V}_t\|_{\mathrm{op}} \leq 1$ and $\|\boldsymbol{x}_t\|_2 \leq 1$, we can show

$$\max_{1 \leq t \leq n} \boldsymbol{z}_t^\top \mathbf{V}_t \boldsymbol{z}_t = \max_{1 \leq t \leq n} \boldsymbol{x}_t^\top \boldsymbol{\Sigma}_{t-1}^{-\frac{1}{2}} \mathbf{V}_t \boldsymbol{\Sigma}_{t-1}^{-\frac{1}{2}} \boldsymbol{x}_t = o_p(1). \tag{64}$$

Besides, equation (61) together with equation (64) implies

$$\max_{1 \leq t \leq n} \boldsymbol{z}_t^\top \mathbf{V}_{t-1} \boldsymbol{z}_t = \max_{1 \leq t \leq n} \frac{\boldsymbol{z}_t^\top \mathbf{V}_t \boldsymbol{z}_t}{1 - \boldsymbol{z}_t^\top \mathbf{V}_t \boldsymbol{z}_t} = o_p(1). \tag{65}$$

Thus, it follows that

$$\begin{aligned}
\max_{1 \leq t \leq n} \|\boldsymbol{w}_t\|_2 &= \max_{1 \leq t \leq n} \left\| \sqrt{1 + \boldsymbol{z}_t^\top \mathbf{V}_{t-1} \boldsymbol{z}_t} \cdot \mathbf{V}_t \boldsymbol{z}_t \right\|_2 \\
&\leq \max_{1 \leq t \leq n} \left( \sqrt{1 + \boldsymbol{z}_t^\top \mathbf{V}_{t-1} \boldsymbol{z}_t} \cdot \|\mathbf{V}_t^{\frac{1}{2}}\|_{\mathrm{op}} \cdot \|\mathbf{V}_t^{\frac{1}{2}} \boldsymbol{z}_t\|_2 \right) \\
&\leq \sqrt{\left( 1 + \max_{1 \leq t \leq n} \boldsymbol{z}_t^\top \mathbf{V}_{t-1} \boldsymbol{z}_t \right) \cdot \max_{1 \leq t \leq n} \boldsymbol{z}_t^\top \mathbf{V}_t \boldsymbol{z}_t} = o_p(1).
\end{aligned} \tag{66}$$

Combining equations (66) and (63) yields (6). Hence we complete the proof by applying Proposition 2.1.

**Remark A.3.** *The detailed proof of Lemma 3.7 can be found in the proof of Theorem 3.6.*

# B   Generalized Theorem 3.6

In Theorem 3.6, we impose the following condition (67) so that the ALEE estimator with weights specified in equation (28) achieves asymptotic normality:

$$\|\mathbf{V}_n\|_{\mathrm{op}} = o_p(1). \tag{67}$$

However, it is typically difficult to directly verify the above condition in practice. To tackle this problem, in this section, we provide a modified version of ALEE estimator which achieves asymptotic normality without requiring condition (67). In this section, we use the same notations $\boldsymbol{\Sigma}_t$, $\boldsymbol{z}_t$, $\mathbf{V}_t$, and $\boldsymbol{w}_t$ as defined in equations (26), (27) and (28), respectively. Furthermore, we let $\lambda_1 \geq \ldots \geq \lambda_n$ be the eigenvalues of the matrix $\mathbf{V}_n^{-1}$ and $\boldsymbol{a}_1, \ldots, \boldsymbol{a}_n$ be the corresponding eigenvectors.

At a high level, we construct additional $m_n$ vectors $\{\boldsymbol{z}_t\}_{n+1 \leq t \leq n+m_n}$ so that the minimum eigenvalue of the resulting matrix $\mathbf{V}_{n+m_n}^{-1}$ is greater than a pre-specified constant $\kappa_n$, which satisfies $\lim_{n \to \infty} \kappa_n = \infty$. It is easy to see that by construction (see Algorithm 1), the matrix $\mathbf{V}_{n+m_n}$ satisfies

$$\|\mathbf{V}_{n+m_n}\|_{\mathrm{op}} \leq \frac{1}{\kappa_n} \xrightarrow{p} 0 \qquad \text{where} \quad m_n = \sum_{k=1}^{d} n_k. \tag{69}$$

**Remark B.1.** *Parameter $\kappa_n$ is set to ensure condition (69) holds. In practice, we set $\kappa_n = d \log(n)$.*

**Remark B.2.** *It's worth mentioning that the number of extra $\{\boldsymbol{z}_t\}_{t>n}$ is a random variable. Therefore, in order to prove a similar asymptotic normality theorem to Theorem 3.6, we have to apply martingale central limit theorem with stopping times [11, Theorem 2.1].*

**Theorem B.3** (Theorem 2.1 in [11])**.** *Let $\{\xi_{n,k}\}_{k \geq 1, n \geq 1}$ be an array of random variables defined on a probability space $(\Omega, \mathcal{F}, P)$ and let $\{\mathcal{F}_{n,k}\}_{n \geq 1, k \geq 0}$ be an array of $\sigma$-fields such that $\xi_{n,k}$ is $\mathcal{F}_{n,k}$-measurable and $\mathcal{F}_{n,k-1} \subset \mathcal{F}_{n,k} \subset \mathcal{F}$ for each $n$ and $k \geqslant 1$. For each $n$, let $k_n$ be a stopping time with respect to $\{\mathcal{F}_{n,k}\}_{k \geq 0}$. Suppose that*

---

**Algorithm 1:** Modified ALEE estimate

---

1: Input:$\{(\boldsymbol{x}_t, y_t)\}_{t=1}^n$ and tuning parameter $\kappa_n$
2: Compute $\{(\boldsymbol{z}_t, \mathbf{V}_t, \boldsymbol{w}_t)\}_{t=1}^n$, $\{(\lambda_k, \boldsymbol{a}_k)\}_{k=1}^d$, and obtain a consistent estimate $\widehat{\sigma}^2$ of $\sigma^2$
3: Initiate $t = n$ and set $\tau_n = 1/\|\boldsymbol{\Sigma}_0^{-1/2}\|_{\mathrm{op}}$
4: **for** $k = 1, \ldots, d$ **do**
5:     Compute $n_k = \lceil \max\{\kappa_n - \lambda_k, 0\} \cdot \tau_n \rceil$
6:     **if** $n_k > 0$ **then**
7:         **for** $i = 1, \ldots, n_k$ **do**
8:             Set $t = t + 1$
9:             Simulate $\epsilon_t \sim \mathcal{N}(0, \widehat{\sigma}^2)$
10:            Define $\boldsymbol{z}_t = \boldsymbol{a}_k/\tau_n$
11:            Compute

$$\mathbf{V}_t = \mathbf{V}_{t-1} - \frac{\mathbf{V}_{t-1}\boldsymbol{z}_t\boldsymbol{z}_t^\top\mathbf{V}_{t-1}}{1 + \boldsymbol{z}_t^\top\mathbf{V}_{t-1}\boldsymbol{z}_t} \quad \text{and} \quad \boldsymbol{w}_t = \sqrt{1 + \boldsymbol{z}_t^\top\mathbf{V}_{t-1}\boldsymbol{z}_t} \cdot \mathbf{V}_t\boldsymbol{z}_t$$

12:         **end for**
13:     **end if**
14: **end for**
15: Obtain $\widehat{\boldsymbol{\theta}}_{\mathrm{ALEE}}$ from equation

$$\sum_{i=1}^n \boldsymbol{w}_i(y_i - \boldsymbol{x}_i^\top\widehat{\boldsymbol{\theta}}_{\mathrm{ALEE}}) + \sum_{i=n+1}^t \boldsymbol{w}_i\epsilon_i = 0 \tag{68}$$

16: Output: $\widehat{\boldsymbol{\theta}}_{\mathrm{ALEE}}$

---

$$\sum_{k=1}^{k_n} \mathbb{E}\left[\xi_{n,k} \mid \mathcal{F}_{n,k-1}\right] \xrightarrow{p} 0, \tag{70a}$$

$$\sum_{k=1}^{k_n} \mathrm{Var}\left[\xi_{n,k} \mid \mathcal{F}_{n,k-1}\right] \xrightarrow{p} 1, \tag{70b}$$

$$\sum_{k=1}^{k_n} \mathbb{E}\left[|\xi_{n,k}|^{2+\delta} \mid \mathcal{F}_{n,k-1}\right] \xrightarrow{p} 0 \quad \textit{for some } \delta > 0, \tag{70c}$$

then $\sum_{k=1}^{k_n} \xi_{n,k} \xrightarrow{d} \mathcal{N}(0, 1)$.

With this setup, we are now ready to prove the asymptotic normality of $\widehat{\boldsymbol{\theta}}_{\mathrm{ALEE}}$ from (68).

**Theorem B.4** (Generalized Theorem 3.6). *Suppose condition* (3) *holds. Then, for any tuning parameters* $\boldsymbol{\Sigma}_0$ *and* $\kappa_n$ *that satisfy* $\|\boldsymbol{\Sigma}_0^{-1}\|_{\mathrm{op}} = o_p(1)$ *and* $\lim_{n\to\infty} \kappa_n = \infty$, *the ALEE estimator* $\widehat{\boldsymbol{\theta}}_{\mathrm{ALEE}}$ *obtained from equation* (68) *satisfies*

$$\left(\sum_{t=1}^n \boldsymbol{w}_t\boldsymbol{x}_t\right) \cdot \frac{\widehat{\boldsymbol{\theta}}_{\mathrm{ALEE}} - \boldsymbol{\theta}^*}{\widehat{\sigma}} \xrightarrow{d} \mathcal{N}(\mathbf{0}, \mathbf{I}_d),$$

*where* $\widehat{\sigma}$ *is a consistent estimator of* $\sigma$.

**Remark B.5.** *We would like to reiterate that the asymptotic variance of of the modified ALEE estimator obtained from* (68) *is the same as the one mentioned in Theorem* 3.6. *Additionally, this modified version does not require the condition* $\|\mathbf{V}_n\|_{\mathrm{op}} = o_p(1)$ *hold and hence is more applicable in practice with theoretical guarantee.*

*Proof.* Rewriting equation (68), we have

$$\sum_{t=1}^n \boldsymbol{w}_t\boldsymbol{x}_t^\top(\widehat{\boldsymbol{\theta}}_{\mathrm{ALEE}} - \boldsymbol{\theta}^*) = \sum_{t=1}^{n+m_n} \boldsymbol{w}_t\epsilon_t. \tag{71}$$

Therefore, by Cramér–Wold theorem, it suffices to show that for any unit vector $\boldsymbol{v}$,

$$\sum_{t=1}^{n+m_n} \boldsymbol{v}^\top \boldsymbol{w}_t \epsilon_t \xrightarrow{d} \mathcal{N}(0, \sigma^2). \tag{72}$$

The proof now follows by verifying the conditions (70a)-(70c) of Theorem B.3 with $\xi_{n,k} = \boldsymbol{v}^\top \boldsymbol{w}_k \epsilon_k$. We begin by verifying conditions (70a)-(70c). By Lemma 3.7, we have

$$\sum_{t=1}^{n+m_n} \boldsymbol{w}_t \boldsymbol{w}_t^\top = \mathbf{I}_d - \mathbf{V}_{n+m_n}. \tag{73}$$

Note that

$$\sum_{t=1}^{n+m_n} \mathrm{Var}[\boldsymbol{w}_t \epsilon_t \mid \mathcal{F}_{t-1}] = \sum_{t=1}^{n+m_n} \sigma^2 \boldsymbol{w}_t \boldsymbol{w}_t^\top + \sigma^2 \left( \frac{\widehat{\sigma}^2}{\sigma^2} - 1 \right) \sum_{t=n+1}^{n+m_n} \boldsymbol{w}_t \boldsymbol{w}_t^\top. \tag{74}$$

By equation (73) and the fact that $\widehat{\sigma}^2$ is consistent, we have

$$\sum_{t=n+1}^{n+m_n} \boldsymbol{w}_t \boldsymbol{w}_t^\top \preceq \mathbf{I}_d \qquad \text{and} \qquad \frac{\widehat{\sigma}^2}{\sigma^2} - 1 \xrightarrow{p} 0. \tag{75}$$

Combining equations (69), (73), (74) and (75), we conclude

$$\sum_{t=1}^{n+m_n} \mathrm{Var}[\boldsymbol{w}_t \epsilon_t \mid \mathcal{F}_{t-1}] \xrightarrow{p} \sigma^2 \mathbf{I}_d. \tag{76}$$

On the other hand, we have

$$\max_{1 \le t \le n+m_n} \|\boldsymbol{w}_t\|_2 \overset{(i)}{\le} \max_{1 \le t \le n+m_n} \left( \sqrt{1 + \boldsymbol{z}_t^\top \mathbf{V}_{t-1} \boldsymbol{z}_t} \cdot \|\mathbf{V}_t\|_{\mathrm{op}} \cdot \|\boldsymbol{z}_t\|_2 \right)$$
$$\overset{(ii)}{\le} \max_{1 \le t \le n+m_n} \sqrt{2} \|\boldsymbol{z}_t\|_2$$
$$\overset{(iii)}{\le} \sqrt{2} \|\boldsymbol{\Sigma}_0^{-1/2}\|_{\mathrm{op}}.$$

Inequality $(i)$ follows from the definition of $\boldsymbol{w}_t$. In inequality $(ii)$, we use the assumption that $\boldsymbol{\Sigma}_0 \succeq \mathbf{I}_d$ and the fact that $\|\boldsymbol{z}_t\|_2 \le 1$ and $\|\mathbf{V}_t\|_{\mathrm{op}} \le 1$. The last inequality $(iii)$ follows from the definition of $\boldsymbol{z}_t$ and the condition that $\|\boldsymbol{\Sigma}_0^{-1}\|_{\mathrm{op}} = o_p(1)$. Hence, we can see that

$$\max_{1 \le t \le n+m_n} \|\boldsymbol{w}_t\|_2 \xrightarrow{p} 0. \tag{77}$$

Therefore, we have

$$\max_{1 \le t \le n+m_n} |\boldsymbol{v}^\top \boldsymbol{w}_t| \xrightarrow{p} 0 \qquad \text{and} \qquad \sum_{t=1}^{n+m_n} \mathrm{Var}[\boldsymbol{v}^\top \boldsymbol{w}_t \epsilon_t \mid \mathcal{F}_{t-1}] \xrightarrow{p} \sigma^2. \tag{78}$$

Note that condition (70a) holds because $\{\boldsymbol{v}^\top \boldsymbol{w}_k \epsilon_k\}_{k \ge 1}$ is a martingale difference sequence by construction. Condition (70b) follows from statement (78). It remains to verify condition (70c). Observe that

$$\sum_{t=1}^{n+m_n} \mathbb{E}[|\boldsymbol{v}^\top \boldsymbol{w}_t \epsilon_t|^{2+\delta} \mid \mathcal{F}_{t-1}] = \sum_{t=1}^{n+m_n} |\boldsymbol{v}^\top \boldsymbol{w}_t|^{2+\delta} \mathbb{E}[|\epsilon_t|^{2+\delta} \mid \mathcal{F}_{t-1}]$$

$$\le \left( \max_{1 \le t \le n+m_n} |\boldsymbol{v}^\top \boldsymbol{w}_t|^{\delta} \right) \cdot \left( \sup_{t \ge 1} \mathbb{E}[|\epsilon_t|^{2+\delta} \mid \mathcal{F}_{t-1}] \right) \cdot \max\{\frac{1}{\sigma^2}, \frac{1}{\widehat{\sigma}^2}\} \sum_{t=1}^{n+m_n} \mathrm{Var}[\boldsymbol{v}^\top \boldsymbol{w}_t \epsilon_t \mid \mathcal{F}_{t-1}]$$

$$\overset{(iv)}{=} o_p(1) \cdot O_p(1) \cdot O_p(1) = o_p(1).$$

Equation $(iv)$ follows from condition (3), equation (78) and the fact that $\widehat{\sigma}^2$ is a consistent estimator. Lastly, by applying Slutsky's theorem, we prove that

$$\frac{1}{\widehat{\sigma}} \sum_{t=1}^{n} \boldsymbol{w}_t \boldsymbol{x}_t^\top (\widehat{\boldsymbol{\theta}}_{\mathrm{ALEE}} - \theta^*) \xrightarrow{d} \mathcal{N}(\mathbf{0}, \mathbf{I}_d). \tag{79}$$

$\square$

# C Simulation

In this section, we provide additional comparisons among the ALEE method, the OLS, the W-decorrelation [8], and the concentration inequality based bounds [1]. The code can be found at https://github.com/mufangying/ALEE.

## C.1 Simulation details

Throughout our experiments, we utilize $\widehat{\sigma}^2$ from equation (9) as an (consistent) estimate of of $\sigma^2$ [19].

**OLS:** When data are i.i.d, the least squares estimator satisfies the following condition

$$\frac{1}{\sigma^2}(\widehat{\boldsymbol{\theta}}_{\mathrm{LS}} - \boldsymbol{\theta}^*)^\top \mathbf{S}_n (\widehat{\boldsymbol{\theta}}_{\mathrm{LS}} - \boldsymbol{\theta}^*) \xrightarrow{d} \chi_d^2.$$

Therefore, we consider $1 - \alpha$ confidence region to be

$$\mathbf{C}_{\mathrm{LS}} = \left\{ \boldsymbol{\theta} \in \mathbb{R}^d : \frac{1}{\widehat{\sigma}^2}(\widehat{\boldsymbol{\theta}}_{\mathrm{LS}} - \boldsymbol{\theta})^\top \mathbf{S}_n (\widehat{\boldsymbol{\theta}}_{\mathrm{LS}} - \boldsymbol{\theta}) \le \chi_{d,1-\alpha}^2 \right\}. \tag{80}$$

We point out that the above confidence region is not guaranteed to be accurate when the data is collected in an adaptive manner, as will also be highlighted in our experiments.

**W-decorrelation:** The W-decorrelation method is borrowed from Algorithm 1 in [8]. Specifically, the estimator takes the form

$$\widehat{\boldsymbol{\theta}}_{\mathrm{W}} = \widehat{\boldsymbol{\theta}}_{\mathrm{LS}} + \sum_{t=1}^n \boldsymbol{w}_t (y_t - \boldsymbol{x}_t^\top \widehat{\boldsymbol{\theta}}_{\mathrm{LS}}). \tag{81}$$

Given a parameter $\lambda$, weights $\{\boldsymbol{w}_t\}_{1 \le t \le n}$ are set as follows

$$\boldsymbol{w}_t = \left( \mathbf{I}_d - \sum_{i=1}^{t-1} \boldsymbol{w}_t \boldsymbol{x}_t^\top \right) \boldsymbol{x}_t / (\lambda + \|\boldsymbol{x}_t\|_2^2). \tag{82}$$

Following the recommendations from the paper [8], in order to set $\lambda$ appropriately, we first run the bandit algorithm or time series with $N$ replications and record the corresponding minimum eigenvalues $\{\lambda_{\min}(\mathbf{S}_n^{(1)}), \ldots, \lambda_{\min}(\mathbf{S}_n^{(N)})\}$. We choose $\lambda$ to be the 0.1-quantile of $\{\lambda_{\min}(\mathbf{S}_n^{(1)}), \ldots, \lambda_{\min}(\mathbf{S}_n^{(N)})\}$. Finally, we obtain a $1 - \alpha$ confidence region for $\boldsymbol{\theta}^*$ as

$$\mathbf{C}_{\mathrm{W}} = \left\{ \boldsymbol{\theta} \in \mathbb{R}^d : \frac{1}{\widehat{\sigma}^2}(\widehat{\boldsymbol{\theta}}_{\mathrm{W}} - \boldsymbol{\theta})^\top \mathbf{W}^\top \mathbf{W} (\widehat{\boldsymbol{\theta}}_{\mathrm{W}} - \boldsymbol{\theta}) \le \chi_{d,1-\alpha}^2 \right\}, \tag{83}$$

where $\mathbf{W} = (\boldsymbol{w}_1, \ldots, \boldsymbol{w}_n)^\top$.

**Concentration based on self-normalized martingales:** We consider [1, Theorem 1] for a single coordinate in two-armed bandit problem and AR(1) model. For contextual bandits, we apply [1, Theorem 2]. Applying concentration bounds requires a sub-Gaussian parameter, for which we use $\widehat{\sigma}$ from equation (9) as an estimate. We point out that this estimate of the sub-Gaussian parameter is conservative, as the sub-Gaussian parameter of a sub-Gaussian random variable is always lower bounded by its variance [33, Chapter 2]. This variance estimate is accurate for Gaussian noise random variables.

- For one dimensional examples, we have that for any $\lambda > 0$, with probability at least $1 - \alpha$:

$$|\widehat{\theta}_{\mathrm{LS}} - \theta^*| \le \frac{\widehat{\sigma}\sqrt{\lambda + \sum_{t=1}^n x_t^2}}{\sum_{t=1}^n x_t^2} \sqrt{\log\left(\frac{\lambda + \sum_{t=1}^n x_t^2}{\lambda \alpha^2}\right)}. \tag{84}$$

  In two-armed bandit problem, $x_t$ is simply $x_{t,1}$ for $\theta_1^*$ or $x_{t,2}$ for $\theta_2^*$. Here we consider $\lambda = 1$.

- For the contextual bandit examples, we apply Theorem 2 from [1], and set $S = \sqrt{d}$; we set a small value of $\lambda = 0.01$ to mimic the performance of an OLS estimators. Specifically, we utilize the following $1 - \alpha$ confidence region

$$\mathbf{C}_{\mathrm{con}} = \left\{ \boldsymbol{\theta} \in \mathbb{R}^d : (\widehat{\boldsymbol{\theta}}_r - \boldsymbol{\theta})^\top (\lambda \mathbf{I}_d + \mathbf{S}_n)(\widehat{\boldsymbol{\theta}}_r - \boldsymbol{\theta}) \le \left( \widehat{\sigma} \sqrt{\log\left(\frac{\det(\lambda \mathbf{I}_d + \mathbf{S}_n)}{\lambda^d \alpha^2}\right)} + \lambda^{\frac{1}{2}} S \right)^2 \right\}, \tag{85}$$

  where $\widehat{\boldsymbol{\theta}}_r = (\mathbf{X}_n^\top \mathbf{X}_n + \lambda \mathbf{I}_d)^{-1} \mathbf{X}_n^\top \mathbf{Y}_n$ and $\mathbf{Y}_n = (y_1, \ldots, y_n)^\top$.

## C.2 Tables for contextual bandits

In all the contextual bandit simulations, we consider noises that are generated from a centered Poisson distribution (i.e. $Poisson(1) - 1$). We would like to highlight that the centered Poisson random variable is not sub-Gaussian. Therefore, it is important to note that concentration inequality-based bounds [1] may not be guaranteed to work. In the simulations of this section, we set the number of samples as $n = 1000$, and the tables below show results over 1000 replications. The tables below clearly show that the average log-volume of the confidence regions are smallest for ALEE among methods which yield valid confidence regions (empirical coverage is more than the target coverage). The volume of the confidence region obtained from the OLS estimate is the smallest, but they under-cover the true parameter. The confidence regions for ALEE are obtained from Theorem B.4 with $\mathbf{\Sigma_0} = \log(n) \cdot \mathbf{I}_d$ and $\kappa_n = d \log(n)$.

Table 2: Contextual bandit: d = 10

| Method | Level of confidence | | | | | |
|---|---|---|---|---|---|---|
| | 0.8 | | 0.85 | | 0.9 | |
| | Avg. Coverage | Avg. log(Volumn) | Avg. Coverage | Avg. log(Volumn) | Avg. Coverage | Avg. log(Volumn) |
| ALEE | 0.819 ($\pm$ 0.385) | -2.761 ($\pm$ 0.263) | 0.872 ($\pm$ 0.334) | -2.370 ($\pm$ 0.263) | 0.920 ($\pm$ 0.271) | -1.894 ($\pm$ 0.263) |
| OLS | 0.807 ($\pm$ 0.395) | -7.306 ($\pm$ 0.262) | 0.863 ($\pm$ 0.344) | -6.915 ($\pm$ 0.262) | 0.905 ($\pm$ 0.293) | -6.439 ($\pm$ 0.262) |
| W-Decorrelation | 0.785 ($\pm$ 0.411) | 8.382 ($\pm$ 0.252) | 0.827 ($\pm$ 0.378) | 8.773 ($\pm$ 0.252) | 0.868 ($\pm$ 0.338) | 9.249 ($\pm$ 0.252) |
| Concentration | 1.000 ($\pm$ 0.000) | 2.517 ($\pm$ 0.252) | 1.000 ($\pm$ 0.000) | 2.548 ($\pm$ 0.252) | 1.000 ($\pm$ 0.000) | 2.591 ($\pm$ 0.252) |

Table 3: Contextual bandit: d = 50

| Method | Level of confidence | | | | | |
|---|---|---|---|---|---|---|
| | 0.8 | | 0.85 | | 0.9 | |
| | Avg. Coverage | Avg. log(Volumn) | Avg. Coverage | Avg. log(Volumn) | Avg. Coverage | Avg. log(Volumn) |
| ALEE | 0.744 ($\pm$ 0.436) | 72.759 ($\pm$ 1.403) | 0.809 ($\pm$ 0.393) | 73.680 ($\pm$ 1.403) | 0.875 ($\pm$ 0.331) | 74.822 ($\pm$ 1.403) |
| OLS | 0.730 ($\pm$ 0.444) | 44.640 ($\pm$ 1.370) | 0.791 ($\pm$ 0.407) | 45.560 ($\pm$ 1.370) | 0.847 ($\pm$ 0.360) | 46.703 ($\pm$ 1.370) |
| W-Decorrelation | 0.192 ($\pm$ 0.394) | 97.559 ($\pm$ 1.337) | 0.225 ($\pm$ 0.418) | 98.479 ($\pm$ 1.337) | 0.276 ($\pm$ 0.447) | 99.622 ($\pm$ 1.337) |
| Concentration | 1.000 ($\pm$ 0.000) | 90.964 ($\pm$ 1.312) | 1.000 ($\pm$ 0.000) | 91.004 ($\pm$ 1.312) | 1.000 ($\pm$ 0.000) | 91.060 ($\pm$ 1.312) |

## C.3 Asymptotic normality with centered Poisson noise variables

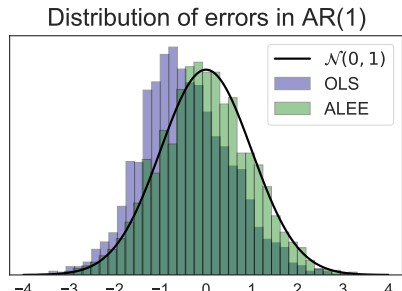
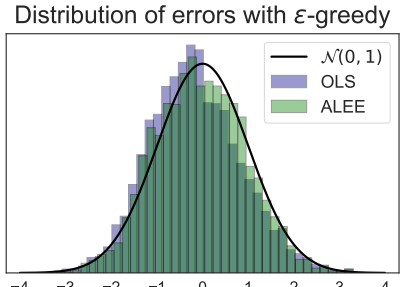

Figure 4: Same setting as Figure 1 but with noise variables $\{\epsilon_t\}$ distributed as centered $Poisson(1)$. We set $n = 3000$ and the number of replications is set to 1000. The simulations show that the asymptotic distribution of ALEE is in good accordance with the asymptotic normality proved in Corollary 3.5 and Theorem 3.1.

