# OpenReview forum: "Adaptive Linear Estimating Equations"
_NeurIPS.cc/2023/Conference — NeurIPS 2023 poster_

### Official Review · Reviewer_G4km · 2023-07-06

**Soundness:** 3 good
**Presentation:** 3 good
**Contribution:** 3 good
**Rating:** 6
**Confidence:** 3

**Summary:**

The authors propose a general method for constructing debiased estimator called Adaptive Linear Estimating Equations (ALEE) estimator, which achieves asymptotic normality even in sequential data collection.

To obtain valid statistical inference, the online debiasing concept is used. The online debiasing procedure guarantees asymptotic reduction of bias to zero, but the convergence speed is slow. However, the ALEE estimator solves the slowly decreasing bias problem.

In this paper, the stable weights of ALEE are proved through equations for three cases: multi-arm bandits, autoregressive time series, and contextual bandits. A comparison of the ALEE method confirms that it performs better than the previous method.

**Strengths:**

- ALEE provides point and interval estimation based on the central limit theorem, which enables stable estimation. It also prove the stability condition of ALEE through formulas for stable weights in three cases (Multi-arm bandits, Autoregressive time series, and Contextual bandits), which increases the efficiency of ALEE. Based on this stability and efficiency, we expect that it can be applied to various models.

- ALEE's training algorithm has fast convergence. The authors provide theoretical performance guarantees and demonstrate ALEE's effectiveness on distributed time series forecasting problems with several examples.


**Weaknesses:**

- In the numerical experiments, the results are only shown using parameter values of 0.3 and 1 for the two-armed bandit setting and the contextual bandit setting. It would be better if used the various values for the parameters.

- Lack of explanation for Table 1.


**Questions:**

Table 1: OLEE typo (ALEE), and Table 1 seems to lack explanation.


**Limitations:**

-

---

> ### Author Rebuttal · Authors · 2023-08-03
>
> We would like to express our sincere appreciation to the reviewer for dedicating the time to reviewing our paper and offering us valuable feedbacks. We truly appreciate your comments and suggestions, and believe they can make our work better.
>
> In the following, we are going to take this opportunity to address each of the points raised in your review, in the order in which they were presented.
>
> **Weakness:**
>
> 1. We performed additional simulations across multiple dimensions and parameter values, incorporating different distributions for noise variables such as mean-centered Poisson, Gamma, and Bernoulli. The outcomes obtained under these settings align with the findings already presented in the paper. We intend to incorporate these simulations into the final version of the paper, as we believe that these additional numerical validations will further strengthen the appeal of the paper. We are grateful to the reviewer for offering this valuable suggestion.
>
> 2. Thank you for bringing this up. We are going to add more explanations of Table 1 in the final version.
>
> **Questions:**
>
> Thank you for catching the typos. We will correct it in the final version.

---

### Official Review · Reviewer_M8Ec · 2023-07-06

**Soundness:** 3 good
**Presentation:** 2 fair
**Contribution:** 3 good
**Rating:** 6
**Confidence:** 2

**Summary:**

This paper considers the problem of least squares when the data is collected sequentially. It proposes a form of weighted least squares where the weights are designed to lead to estimates that are asymptotically normal and nearly optimal variance. The appropriate weights are derived for the multi-arm bandit, autoregressive, and context bandit settings. Experimental results on some toy datasets confirm the theory.

**Strengths:**

1. The proposed estimator is simple and nearly efficient.
2. Therefore, I think it would be useful for practitioners.
3. The problem is well-motivated.
I am not familiar with the literature on this problem, so I cannot comment on originality.

**Weaknesses:**

1. The presentation is somewhat confusing at times. The matrix $A$ from Section 2 does not appear in Section 3. The general construction strategy in Section 3.1 does not seem to be applied in Section 3.3 and it's not clear why.
2. The examples used in the experiments are all very simple. It would strengthen the work to, for example, vary the number of arms/dimension.

**Questions:**

Please see above.

**Limitations:**

The paper is largely theoretical, so I think it's fine that the authors don't include a broader impacts section. However, I think the authors should include something about the limitations of the current work (see previous sections).

---

> ### Author Rebuttal · Authors · 2023-08-03
>
> We would like to express our sincere appreciation to the reviewer for dedicating the time to reviewing our paper and offering us valuable feedbacks. We truly appreciate your comments and suggestions, and believe they can make our work better.
>
> In the following, we are going to take this opportunity to address each of the points raised in your review, in the order in which they were presented.
>
> **Weakness:**
>
> We apologize for any confusion caused by our presentation, and we would like to clarify certain aspects of the paper to make it clearer.
>
> 1. A benefit of using $A_w$ is that it allows for a better evaluation of the efficiency of the ALEE estimator, and it also serves as motivation for constructing weights $w_t$ with desirable properties (see equations (10) and (11)). We would like to highlight that the matrices $A_w$ and $W^\top X$ yield the same variance asymptotically, meaning that $A_w^\top A_w =   X^\top W W^\top X + \text{ lower order term}$, and consequently, they can be used interchangeably; see the discussion near equations (8) and (9) where we already discuss the utility of using the matrix $A_w$. In section 3 onwards, we use the matrix $W^\top X$ because it is easier to understand. We sincerely appreciate the reviewer for bringing this to our attention, and we will add additional clarification to the paper to highlight the utility of the matrix $A_w$.
>
>  2. We thank the reviewer for this careful observation. In Section 3.1, we consider the multi-armed bandit problem, where the problem is ``effectively" one dimensional due to the orthogonality of the columns of the data matrix $X$. In dimension $d = 1$, we can do nice manipulations to the variance term which allows us to simplify the asymptotic variance term, and the resulting corollaries are much more informative and interpretable.   In Section 3.3,  we tackle the general contextual bandit problem, where the columns of the columns of the data-matrix matrix $X$ are no longer orthogonal, and we had to develop ideas that apply to dimension $d > 1$. This is why the constructions in section 3.1 and section 3.3 --- although they have same goal --- are very different.
>
> 3. We conducted some additional simulations with different noise variables (mean-centered Poisson, Gamma, and Bernoulli noise variables) and for various dimensions and parameter values. The results obtained using these settings align with what we have presented in the paper. In the final version of the paper, we plan to include these simulations. We believe that these additional numerical validations will enhance the appeal of the paper, and we sincerely appreciate the reviewer for providing this valuable suggestion.
>
> **Limitations:** Thank you for your suggestions. We apologize for not including the limitation of the work and will add it in the final version of  the paper.

---

> > ### Comment · Reviewer_M8Ec · 2023-08-11
> > **Thanks to the authors for the rebuttal**
> >
> > After reading it and the other reviews, I have decided to keep the same score.

---

### Official Review · Reviewer_ADJL · 2023-07-21

**Soundness:** 3 good
**Presentation:** 3 good
**Contribution:** 3 good
**Rating:** 7
**Confidence:** 2

**Summary:**

This paper proposes an estimator (ALEE) for adaptively collected data generated from adaptive linear models, describes its construction such that asymptotic normality holds for practically relevant examples, and demonstrates its desirable properties in numerical experiments.

**Strengths:**

I enjoy reading this paper. It reads well.

- The problem of inference for adaptive data is practically relevant
- Theoretical guarantees assume somewhat weaker assumptions than previous works
- The proposed method provides an improvement over other approaches, at least in the numerical experiments shown

**Weaknesses:**

I don't think this submission lacks anything for a NeurIPS paper

**Questions:**

- One novelty of the paper is that Eq (3) is weaker than sub-Gaussian, yet all experiments are for Gaussian noise. I would be more convinced if the numerical simulations shows the same desirable properties for non-Gaussian noise.

- Perhaps refer to Fig 1 from the main text?

Some minor typos I find:

- Refs 14&15 are identical
- l51 - _debaising_ → debiasing
- l98 - _Slutsy's theorem_ → Slutsky's theorem

---

> ### Author Rebuttal · Authors · 2023-08-03
>
> We would like to express our sincere appreciation to the reviewer for dedicating the time to reviewing our paper and offering us valuable feedbacks. We truly appreciate your comments and suggestions, and believe they can make our work better.
>
> In the following, we are going to take this opportunity to address each of the points raised in your review, in the order in which they were presented.
>
> **Strengths and weaknesses:**  we are glad that you enjoy reading this paper and thank you so much for your kind words.
>
> **Questions:**
>
> Thank you for your suggestion. We conducted simulations with mean-centered Poisson, Gamma, and Bernoulli noise variables. The results obtained using these noise variables align with what we have presented in the paper (please see Figure 1 from the paper). In the final version of the paper, we plan to include these simulations. We believe that these additional numerical validations will enhance the appeal of the paper, and we sincerely appreciate the reviewer for providing this valuable suggestion.
>
> Lastly, thank you for catching the typos, we will correct them in the final version.

---

> > ### Comment · Reviewer_ADJL · 2023-08-11
> > **Thanks for the reply**
> >
> > I am satisfied with the response. I am keeping my rating as is.

---

### Official Review · Reviewer_Z7he · 2023-07-26

**Soundness:** 3 good
**Presentation:** 3 good
**Contribution:** 3 good
**Rating:** 5
**Confidence:** 1

**Summary:**

This paper introduces a general method for constructing debiased estimator within the context of sequential data collection. The proposed methodology is applied explicitly to multi-arm bandits, autoregressive time series, and contextual bandits. Experiments are conducted in these three domains to verify the applicability and effectiveness.

**Strengths:**

1.	The proposed method is able to achieve asymptotic normality without knowing the data collection algorithm and can obtain a faster convergence rate of the bias term.
2.	Pointwise and interval estimates can therefore be generated.


**Weaknesses:**

The experiment section only contains synthetic results. Considering the broad application of sequential data collection, it would greatly enhance the study if the authors could validate their framework using real-world datasets. This could provide more practical insight into the effectiveness of the proposed method.

**Questions:**

In Figures 2 and 3, what do “lower tail coverage” and “upper tail coverage” represent?

**Limitations:**

 Limitations are not mentioned, and the potential negative societal impact is not addressed.

---

> ### Author Rebuttal · Authors · 2023-08-03
>
> We would like to express our sincere appreciation to the reviewer for dedicating the time to reviewing our paper and offering us valuable feedbacks. We truly appreciate your comments and suggestions, and believe they can make our work better.
>
> In the following, we are going to take this opportunity to address each of the points raised in your review, in the order in which they were presented.
>
> **Weaknesses:** Thank you for your suggestion, and we agree with the reviewer that incorporating additional numerical/real data simulations will enhance the appeal of the paper. In the current version of the paper, we have primarily focused on simulated experiments. The use of simulated experiments allows us to establish a known ground truth, making it easier to compare the utility of different methods.
> In the revised version of the paper, we plan to supplement the existing simulations with new ones involving different noise variables, such as mean-centered Poisson and Gamma noise variables. These additional results will align with the findings presented in the paper (please refer to Figure 1). Furthermore, we intend to include a real data example and assess the performance of various methods on it. While real data examples lack a known ground truth, comparing the confidence interval widths of different methods will be beneficial in evaluating their effectiveness. We extend our gratitude to the reviewer once again for this excellent suggestion.
>
> **Questions:**  Thank you for your question. Lower tail coverage represents whether the lower one-sided confidence interval, which has the form$(-\infty, a)$, can cover the true parameter. Upper tail coverage corresponds to the upper one-sided confidence interval, which has the form $(a, \infty)$.
>
> **Limitations:**  We apologize for not including the limitation part and not addressing the potential negative societal impact.  We are going to add them in the final version. Thank you for your suggestion.

---

> > ### Comment · Reviewer_Z7he · 2023-08-19
> > **Thanks for the Authors' response**
> >
> > The authors' response is promising to me. I will keep my score as it is.

---

### Author Rebuttal · Authors · 2023-08-10

Due to the page limit, we only include part of the updated simulations. Please see attached pdf file for simulations regarding different distributions for the noise variables and varying dimension $d$ for the contextual bandit example.

We plan to add limitation in the discussion section.

**Limitations:** In our paper, we propose ALEE estimator that achieves asymptotic normality and discuss its optimality. Our result is based on large samples and it would be interesting to investigate how to improve the efficiency of ALEE framework under finite samples.

---

### Decision · Program_Chairs · 2023-09-21

**Decision:**

Accept (poster)

**Comment:**

Statistical inference on data collected by sequential data collection algorithms such as bandit algorithms is a topic of much contemporary relevance and interest. This paper considers an adaptive linear model and proposes an adaptive-linear-estimation-equations based approach that corrects for the bias in the ordinary least squares (OLS) solution. It is a very nice case of merging asymptotic normality arguments with finite sample concentration inequalities. All reviewers are unanimous in recommending acceptance.